# Leveraging the histidine kinase-phosphatase duality to sculpt two-component signaling

Stefanie S. M. Meier [1,6], Elina Multamäki [2,6], Américo T. Ranzani [1], Heikki Takala [2,3] ✉ & Andreas Möglich [1,4,5] ✉

Bacteria must constantly probe their environment for rapid adaptation, a crucial need most frequently served by two-component systems (TCS). As one component, sensor histidine kinases (SHK) control the phosphorylation of the second component, the response regulator (RR). Downstream responses hinge on RR phosphorylation and can be highly stringent, acute, and sensitive because SHKs commonly exert both kinase and phosphatase activity. With a bacteriophytochrome TCS as a paradigm, we here interrogate how this catalytic duality underlies signal responses. Derivative systems exhibit tenfold higher red-light sensitivity, owing to an altered kinase-phosphatase balance. Modifications of the linker intervening the SHK sensor and catalytic entities likewise tilt this balance and provide TCSs with inverted output that increases under red light. These TCSs expand synthetic biology and showcase how deliberate perturbations of the kinase-phosphatase duality unlock altered signal-response regimes. Arguably, these aspects equally pertain to the engineering and the natural evolution of TCSs.

Two-component signaling systems (TCS) represent the dominant means by which bacteria sense and respond to their environment[1–3]. In their canonical form, TCSs comprise a membrane-spanning, homodimeric sensor histidine kinase (SHK) that adjusts in signal-dependent manner the phosphorylation status of a second component, the response regulator (RR). SHK autophosphorylation at an eponymous histidine residue precedes phosphotransfer to a conserved aspartate within the RR receiver domain. SHKs commonly feature sensor modules situated in the periplasm or the extracellular space that connect to intracellular effector moieties via transmembrane linkers of α-helical and coiled-coil conformation. Thereby, a conduit between extracellular/periplasmic signals, e.g., the concentration of metabolites, and intracellular physiological responses is established. Certain SHKs are soluble cytosolic receptors, provided they respond to signals that can traverse the membrane, e.g., oxygen[4,5] and light[6], or that arise intracellularly, e.g., the second messenger cyclic di-GMP[7].

SHKs bind and phosphorylate their cognate RRs with high specificity, thus allowing the coexistence in a single cell of multiple TCSs that are well insulated from one another. Upon signal-dependent phosphorylation by the SHK, the RR ushers in cellular responses, frequently by initiating transcription from target promoters. To accomplish particularly stringent and steep responses to signal changes, SHKs commonly possess dual enzymatic activity. Not only do they mediate autophosphorylation and phosphotransfer to the RR, but also, they catalyze the hydrolysis of the phosphorylated RR[8,9]. The downstream physiological response is determined by the levels of phosphorylated RR, which in turn depend on the net balance between the opposing elementary kinase and phosphatase activities. In the absence of signal, one of the two elementary activities usually prevails, and the SHK operates as either a net kinase or net phosphatase. Signal modulates the balance between the elementary activities and can shift the SHK from its kinase-active (denoted K state) to a phosphatase-active state (P state), or vice versa.

[1]Department of Biochemistry, University of Bayreuth, Bayreuth, Germany. [2]Department of Anatomy, University of Helsinki, Helsinki, Finland. [3]Department of Biological and Environmental Science, Nanoscience Center, University of Jyvaskyla, Jyvaskyla, Finland. [4]Bayreuth Center for Biochemistry & Molecular Biology, Universität Bayreuth, Bayreuth, Germany. [5]North-Bavarian NMR Center, Universität Bayreuth, Bayreuth, Germany. [6]These authors contributed equally: Stefanie S. M. Meier, Elina Multamäki. ✉e-mail: heikki.p.takala@jyu.fi; andreas.moeglich@uni-bayreuth.de

Certain SHKs may lack either of the two activities as exemplified by the bacteriophytochrome (BphP) from *Deinococcus radiodurans* that is devoid of kinase activity and exclusively acts as a red-light-activated phosphatase[10]. As receptors of red and far-red light, BphPs comprise photosensory modules (PSM) with covalently bound biliverdin chromophores and often control two-component signaling (Fig. 1a)[8,11,12]. Prototypical BphPs adopt in darkness their red light-absorbing (Pr) state. Red light drives the transition to the far-red light-absorbing (Pfr) signaling state, which reverts to Pr either thermally or driven by far-red light[13,14].

An abundance of high-resolution structures of SHKs, both at full length and as truncated fragments[15–21], has provided molecular insight into the modes underpinning the transition between the K and P functional states. Notwithstanding considerable diversity of the SHK systems investigated structurally, overarching principles emerge[1,8,22]. In both the K and P states, the DHp subdomain (dimerization and histidine phosphotransfer) of the SHK generally forms a tetrameric antiparallel coiled coil to which the C-terminal CA domains (catalytic and ATP-binding) are appended laterally via flexible linkers (Suppl. Fig. 1). Within the K state, the CA domains can reach and thereby phosphorylate the phospho-accepting histidine within the DHp domain. Often, the K state exhibits an asymmetric structure which may be instrumental for histidine phosphorylation and subsequent phosphotransfer[1]. By contrast, the P state is usually C2-symmetric with the CA domains and/or the active-site histidine sequestered to prevent SHK autophosphorylation and phosphotransfer to the RR. Rather, residues involved in catalyzing the hydrolysis of the phospho-RR are moved into place. Individual SHKs substantially differ in how the CA domains are kept from catalyzing the autokinase and phosphatase reactions within their P states[7]. Many SHKs are connected to their N-terminal sensor modules by coiled-coil linkers[23] (see, e.g., Suppl. Fig. 1) which undergo signal-dependent conformational changes that underlie the K⇌P switch. Irrespective of the precise nature of these changes, they may be limited in structural extent and associated free energy difference. SHKs are poised between the K and P states, thereby allowing even subtle perturbations, e.g., by introduction of signal or by single-residue alterations[24–27], to profoundly shift their net activities.

Here, we harness a red-light-responsive TCS[28] with BphP input as a paradigm to interrogate how deliberate modifications of the sensor and linker domains reprogram the activity and signal response. The modular exchange of the BphP PSM increases light sensitivity of the TCS although light absorption per se is hardly affected. Rather, the sensor exchange tilts the balance between the elementary kinase and phosphatase activities. This balance can also be controlled by extending and shortening the linker between the input and output entities to give rise to SHK variants with inverted response to red light.

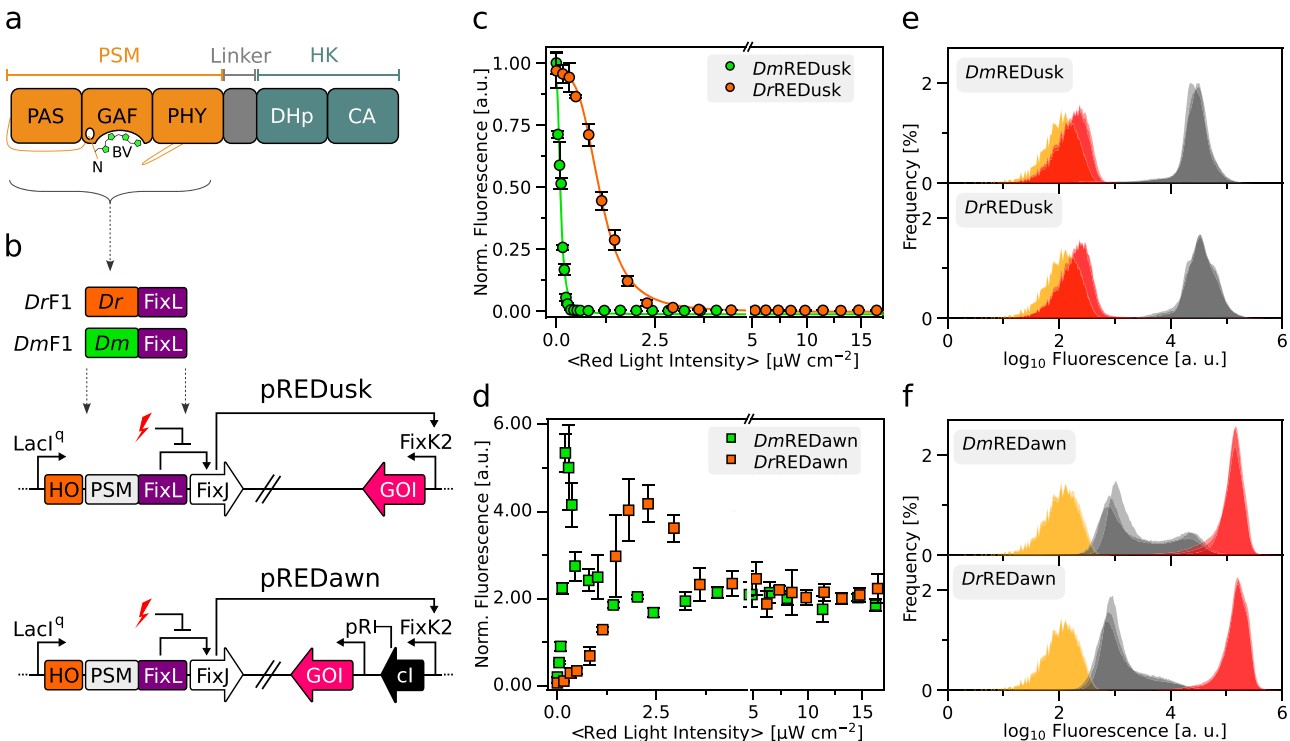

**Fig. 1 | Architecture and characterization of plasmid systems for red-light-controlled gene expression in bacteria. a** Bacteriophytochromes (BphP) possess N-terminal photosensory modules (PSM) which covalently bind biliverdin (BV) chromophores and comprise consecutive PAS, GAF, and PHY domains. As exemplified for the BphP from *Deinococcus radiodurans*, the C-terminal effector module is most often a sensor histidine kinase consisting of DHp (dimerization and histidine phosphotransfer) and CA (catalytic and ATP-binding) domains. **b** In the pREDusk circuits, the LacI$^q$ promoter controls the expression of a tricistronic operon comprising a heme oxygenase (HO), a sensor histidine kinase (SHK) (i.e., *Dr*F1 or *Dm*F1), and the response regulator (RR) FixJ from *Bradyrhizobium japonicum*. *Dm*F1 and *Dr*F1 consist of the PSM from the *Deinococcus maricopensis* and *D. radiodurans* BphPs, respectively, linked to the effector module of the *B. japonicum* FixL SHK. In darkness, *Dm*F1/*Dr*F1 drive FixJ phosphorylation and thereby activate expression of a gene of interest (GOI) from the FixK2 promoter. The pREDawn system extends pREDusk by an additional λ phage cI repressor and its target promoter pR, which induces inversion and amplification of the circuit response to red light. **c** *Ds*Red production of bacteria harboring *Dm*REDusk (green circles) or *Dr*REDusk (orange circles) as a function of red-light intensity. Fluorescence readings were normalized to the optical density of the bacterial cultures at 600 nm ($OD_{600}$) and corrected for background fluorescence. Data represent mean ± s.d. of $n = 3$ biologically independent replicates and were normalized to *Dm*REDusk in darkness. Light intensities are averaged over the duty cycle as marked by angled brackets. **d** As **c** but for *Dm*REDawn (green squares) and *Dr*REDawn (orange squares). **e** Analysis by flow cytometry of bacteria carrying *Dm*REDusk (top) or *Dr*REDusk (bottom) and incubated under red light (red) or in darkness (gray). The yellow curves denote control bacteria harboring empty vectors with a multiple-cloning site replacing *Ds*Red. $n = 3$ biologically independent replicates for each sample are shown. **f** As panel **e** but for *Dm*REDawn and *Dr*REDawn. Source data are provided as a Source Data file.

Resultant SHK variants with enhanced sensitivity and reprogrammed activity could serve as tools for applications in optogenetics[29], synthetic biology, and biotechnology. More generally, they illustrate how exploiting the kinase-phosphatase duality can utterly reprogram the SHK signal response – be it by design or by evolution.

## Results

### Boosting the sensitivity of a red-light-responsive histidine kinase

We recently engineered the chimeric SHK $Dr$F1 as a fusion between the photosensory module (PSM) of the *D. radiodurans* BphP and the DHp/CA effector moiety of *Bradyrhizobium japonicum* FixL (Fig. 1b)[4,28]. Combined with the cognate RR FixJ from *B. japonicum*, $Dr$F1 forms a TCS and affords gene expression from target promoters that can be dialed down by over 200-fold under red light compared to darkness. For applications in optogenetics[29] and biotechnology, all system components were assembled on the portable plasmid pREDusk[28] (Fig. 1b). Using this plasmid as a platform, we explored how replacement of the *D. radiodurans* PSM ($Dr$PSM) for another PSM impacts on TCS output and response to light. We opted for the PSM of a BphP from *Deinococcus maricopensis* ($Dm$PSM) because this module recently proved efficient at regulating an adenylyl cyclase, outperforming around twenty other PSMs, including the $Dr$PSM[30]. Informed by a sequence alignment (Suppl. Fig. 2), we generated the $Dm$F1 SHK by introducing the $Dm$PSM (residues 1-510) in lieu of the $Dr$PSM while keeping the $Bj$FixL fragment (residues 266-505) unaltered (Fig. 1b). Within the pREDusk context, $Dm$F1 drove expression of the red-fluorescent reporter gene *Ds*Red in darkness that was strongly suppressed under red light (650 nm) (Suppl. Fig. 3). In the following, we refer to this setup as $Dm$REDusk, while the original system it derives from[28] will be designated as $Dr$REDusk (Suppl. Table 1).

We next assessed bacteria harboring the $Dm$REDusk circuit in depth and found that the reporter fluorescence monotonically decreased with increasing red-light dose by around 300-fold, comparable to $Dr$REDusk (Fig. 1c). With a half-maximal light dose of $(0.11 \pm 0.01)\,\mu\text{W cm}^{-2}$, $Dm$REDusk intriguingly proved tenfold more light-sensitive than $Dr$REDusk $[(1.11 \pm 0.02)\,\mu\text{W cm}^{-2}]$. It is worth noting that, with certain exceptions[31], the PSMs of BphPs generally exhibit overall quantum yields of around 0.1–0.2 for Pr→Pfr photoconversion[32]. That is, only 10–20% of the absorbed light quanta trigger a downstream signal response. Although no pertinent experimental data are available for the *D. radiodurans* BphP, a recent simulation study puts its quantum yield for Pr→Pfr photoconversion at 15%[33]. Even in the absence of further investigation, it is hence likely that the unexpected and puzzling difference in light sensitivity between $Dm$REDusk and $Dr$REDusk cannot be explained by differing quantum yields for the Pr→Pfr transition, at least not to full extent, as quantum yields generally range between zero and unity.

The striking sensitivity difference also manifested in the pREDawn context that derives from pREDusk and harbors a gene-inversion cassette based on the λ phage cI repressor[34] that inverts and amplifies the response to light (Fig. 1b). When exposed to increasing doses of red light, the *Ds*Red reporter expression in bacteria bearing $Dm$REDawn levelled up by around 26-fold at a light intensity of $0.2\,\mu\text{W cm}^{-2}$, before again dropping threefold at even higher intensities (Fig. 1d). For reference, $Dr$REDawn mediated an increase of *Ds*Red expression by around 61-fold at a light dose of $2.3\,\mu\text{W cm}^{-2}$, which decreased by twofold at higher light intensities. As reported previously[28], the attenuated response at high light intensities likely owes to the presence of the cI repressor and its strong target promoter pR. That notwithstanding, introduction of the $Dm$PSM garnered a 10-fold light sensitivity gain, as it did in the $Dm$REDusk context.

To glean additional insight, we probed the red-light-gated gene-expression circuits by flow cytometry. Under non-inducing conditions, i.e., saturating levels of red light (650 nm, $100\,\mu\text{W cm}^{-2}$), both

$Dm$REDusk and $Dr$REDusk exhibited uniformly low reporter fluorescence that was only 1.6-fold above the background fluorescence of a control circuit bearing a multiple-cloning site (MCS) rather than the fluorescence reporter (Fig. 1e). Transfer to darkness incurred a uniform shift of the single-cell fluorescence distribution to 56-fold higher values. Both the $Dm$REDawn and $Dr$REDawn exhibited some leakiness under non-inducing, dark conditions (Fig. 1f). While the single-cell fluorescence peaked at values around 2.1-fold higher than those of the MCS controls, subpopulations of cells exhibited partial induction with higher reporter fluorescence. We provisionally ascribe these subpopulations to continuous expression of the strong cI repressor which arguably imposes a considerable metabolic burden on the bacteria. Under inducing, red-light conditions, the single-cell fluorescence distributions of $Dm$REDawn and $Dr$REDawn coalesced to uniform populations at around 410-fold and 500-fold higher values, respectively, than the main peak in darkness.

On the whole, the exchange of the $Dr$PSM input module for $Dm$PSM consistently installed higher light sensitivity in both the pREDusk and pREDawn circuits. Higher sensitivity may be advantageous for multiplexing of several optogenetic circuits and for applications where light delivery is limiting.

### Photochemical properties alone cannot explain the enhanced light sensitivity

To unravel the underpinnings of the more sensitive light response in the $Dm$-based vs. the $Dr$-based gene-expression circuits, we expressed and purified the underlying $Dm$F1 and $Dr$F1 photoreceptors. Both receptors covalently incorporated their biliverdin chromophore to similar extents, as gauged by absorbance spectroscopy (Fig. 2a, b, Suppl. Fig. 4). Upon far-red illumination (800 nm), both $Dm$F1 and $Dr$F1 assumed the Pr state, characterized by the so-called Soret and Q absorbance bands at 400 nm and 700 nm, respectively (Fig. 2a, b). Exposure to red light (650 nm) populated the Pfr state with Soret and Q band maxima at 410 and 752 nm, respectively. As red light also drives the reverse Pfr→Pr transition, if to lesser degree, a photostationary state resulted with Pr:Pfr ratios of 0.17:0.83 for $Dm$F1 and 0.09:0.91 for $Dr$F1. Evidently, red light converted $Dr$F1 to its Pfr state to slightly higher, but overall similar extents compared to $Dm$F1.

We next reasoned that $Dm$F1 and $Dr$F1 may well differ in how efficient red light is at driving their Pr→Pfr transition. At a given light intensity, the velocity of this transition scales with the extinction coefficient at the excitation wavelength times the quantum yield for productive photochemistry leading to biliverdin isomerization[11,32]. Notably, $Dm$F1 and $Dr$F1 did not substantially differ in the degree of BV incorporation (Suppl. Fig. 4), and they showed similar absorbance properties (see Fig. 2a, b). Hence, for a given intensity of red light, the kinetics of the Pr→Pfr conversion provide an approximate handle on the relative quantum yields in $Dm$F1 and $Dr$F1. We accordingly recorded photoactivation kinetics by illuminating the sample with red light $(1\,\text{mW cm}^{-2}, 650\,\text{nm})$ while monitoring over time the change of the absorbance signal at 750 nm ($A_{750}$) which reports on the fractional Pfr population. For both $Dm$F1 and $Dr$F1, the monotonic increase of the $A_{750}$ signal could be described by single-exponential functions with rate constants of $0.036\,\text{s}^{-1}$ and $0.028\,\text{s}^{-1}$, respectively (Fig. 2c). The somewhat faster kinetics in $Dm$F1 hint at a more efficient, light-driven Pr→Pfr transition which, all other things being equal, translates into a higher light sensitivity. However, given its magnitude of around 30%, it appears unlikely that this effect alone can account for the tenfold sensitivity difference evidenced above in the $Dm$REDusk and $Dr$REDusk circuits. We also probed the photoactivation kinetics of the reverse Pfr→Pr process under far-red light $(5\,\text{mW cm}^{-2}, 800\,\text{nm})$. Interestingly and for unknown molecular reasons, these kinetics were around 2.7 times faster in $Dm$F1 relative to $Dr$F1, indicative of a more efficient light-driven return to Pr (Fig. 2c). As the above pREDusk reporter-gene experiments were, however, conducted using only red

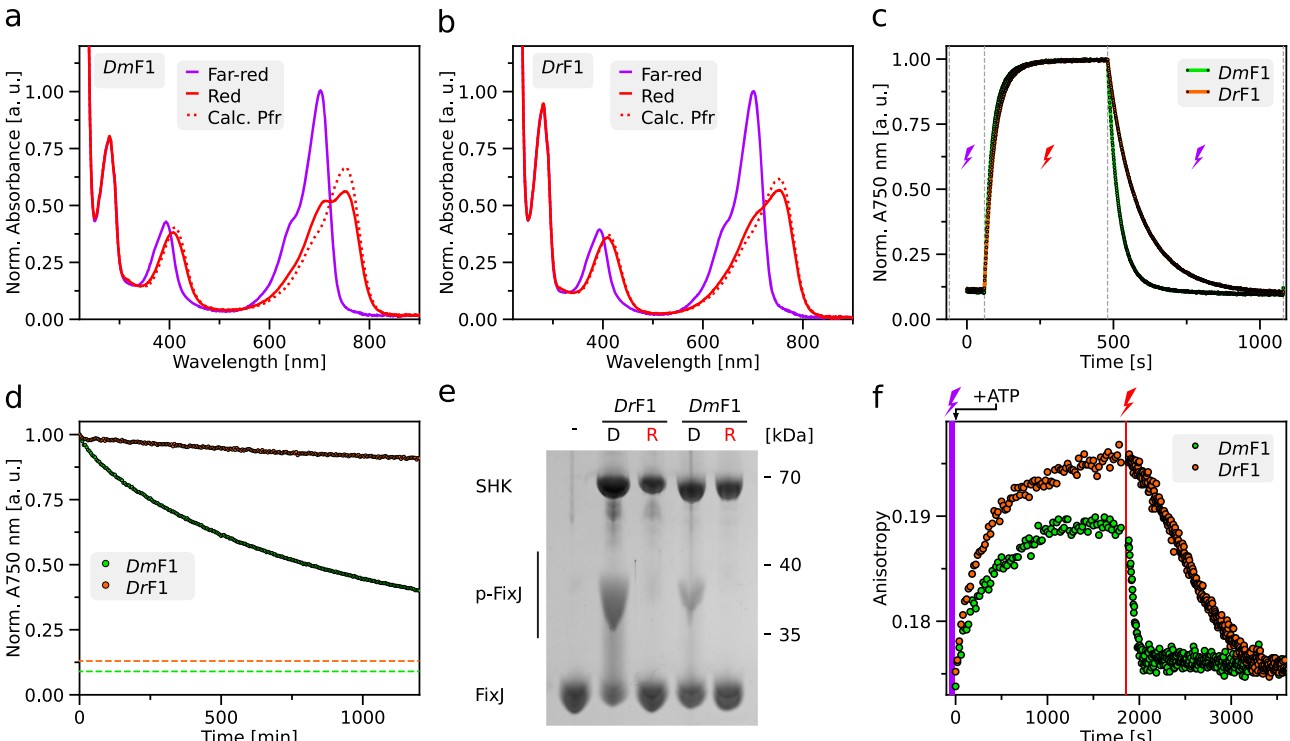

**Fig. 2 | Spectroscopic and enzymatic characterization of *Dm*F1 and *Dr*F1.**
**a** UV–vis absorbance spectra of *Dm*F1 upon illumination with far-red light (800 nm, purple line) or red light (650 nm, red line). The dotted line denotes the pure Pfr spectrum calculated according to[69]. All spectra were normalized to the absorbance at 700 nm after far-red light exposure. **b** As panel **a** but for *Dr*F1. **c** The Pr→Pfr photoactivation kinetics of *Dm*F1 (green) and *Dr*F1 (orange) driven by red light (red flash), and the corresponding Pfr→Pr reversion kinetics under far-red light (purple flash) were monitored by the absorbance at 750 nm and evaluated according to single-exponential functions. **d** The dark recovery of *Dm*F1 (green) and *Dr*F1 (orange) after illumination with saturating red light was observed at 750 nm. For reference, the dashed green and orange lines denote the absorbance at 750 nm in the far-red illuminated spectra of *Dm*F1 and *Dr*F1 (see panels **a**, **b**). **e** When incubated in darkness ('D') in the presence of ATP, both *Dm*F1 and *Dr*F1 trigger an increase in phosphorylated FixJ (p-FixJ) as visualized by slower migration in a Phos-tag gel than the unphosphorylated form. By contrast, under red light ('R'), FixJ remains unphosphorylated, as it does if no SHK is included in the reaction ('-'). Experiments were repeated three times with similar outcomes. **f** *Dm*F1 and *Dr*F1 were incubated in the presence of FixJ and a double-stranded DNA molecule containing the operator site for phospho-FixJ and a 5′-attached TAMRA fluorescence label (see Suppl. Fig. 6a). Following illumination with far-red light (to populate the Pr state, indicated by purple bar), the reaction was initiated by ATP addition, and the TAMRA fluorescence anisotropy was monitored over time. The signal rise indicates formation of phospho-FixJ and binding to the DNA. Illumination with red light (marked in red) leads to a decrease in signal, indicative of net phosphatase activity. Experiments were done in triplicate ($n = 3$) with similar outcomes (see Suppl. Fig. 6b, c). Source data are provided as a Source Data file.

light, this difference cannot explain the observed sensitivity differences either.

The Pfr state may also revert to Pr thermally in the so-called dark recovery reaction. As for instance evidenced in certain blue-light receptors[23,35,36], changes in the dark-recovery kinetics can translate into sizeable sensitivity differences under prolonged illumination. Put simply, if a given photoreceptor recovers to its dark-adapted state much faster than another receptor, it will take proportionally more light to populate the light-adapted signaling state to the same degree at photostationary state[32,37]. To investigate potential differences in dark recovery, *Dr*F1 and *Dm*F1 were illuminated with red light, and the recovery to the Pr state was monitored over time at 25 °C by tracking $A_{750}$ (Fig. 2d). In line with earlier measurements on BphPs[38], the recovery was slow and incomplete even after 24 h for both receptors (Suppl. Figs. 4c and 5). Over the initial 20 h, the recovery proceeded with similar kinetics in *Dm*F1 (0.098 h$^{-1}$) and *Dr*F1 (0.059 h$^{-1}$), but the amplitude was markedly higher in case of *Dm*F1 (66% versus 10%), leading to faster overall dark reversion (Fig. 2d, Suppl. Fig. 5). Longer measurements were precluded because of the onset of protein aggregation. Given its slow kinetics, we expect the dark recovery to have a minor effect on the pREDusk and pREDawn reporter-gene experiments. If anything, the slower and less complete recovery of *Dr*F1 should translate into a higher light sensitivity at photostationary state for *Dr*REDusk relative to *Dm*REDusk but the opposite is the case. Taken together, we therefore conclude that the photochemical properties of *Dm*F1 and *Dr*F1 do not underlie the observed sensitivity difference.

## Molecular basis of enhanced light sensitivity

We suspected that the diverging dose-response profiles of *Dm*REDusk and *Dr*REDusk might be rooted in the enzymatic activities of their underlying *Dm*F1 and *Dr*F1 SHKs under different light conditions. Thus, we assayed the phosphotransfer activity of *Dm*F1 and *Dr*F1 to the FixJ RR using Phos-tag assays[10]. Briefly, these experiments resort to polyacrylamide gels containing a chelating agent that lowers the electrophoretic migration speed of phosphorylated proteins relative to their non-phosphorylated forms. Upon ATP addition, both *Dm*F1 and *Dr*F1 promoted the phosphorylation of FixJ in darkness but not under red light (Fig. 2e). These observations agree with the pREDusk reporter-gene assays and indicate that both *Dm*F1 and *Dr*F1 act as net kinases in darkness. That is, their elementary kinase activity outweighs their phosphatase activity (if present) under these conditions.

To get quantitative insight into these elementary activities of *Dm*F1 and *Dr*F1, we next adapted a histidine-kinase assay pioneered for the blue-light-responsive SHK YF1[23,39]. In brief, the assay employs a double-stranded DNA fragment that harbors the binding site for phospho-FixJ and is labeled with tetramethylrhodamine (TAMRA). Once phosphorylated by an SHK, FixJ binds to the DNA, thereby incurring a decrease in rotational diffusion and a concomitant increase

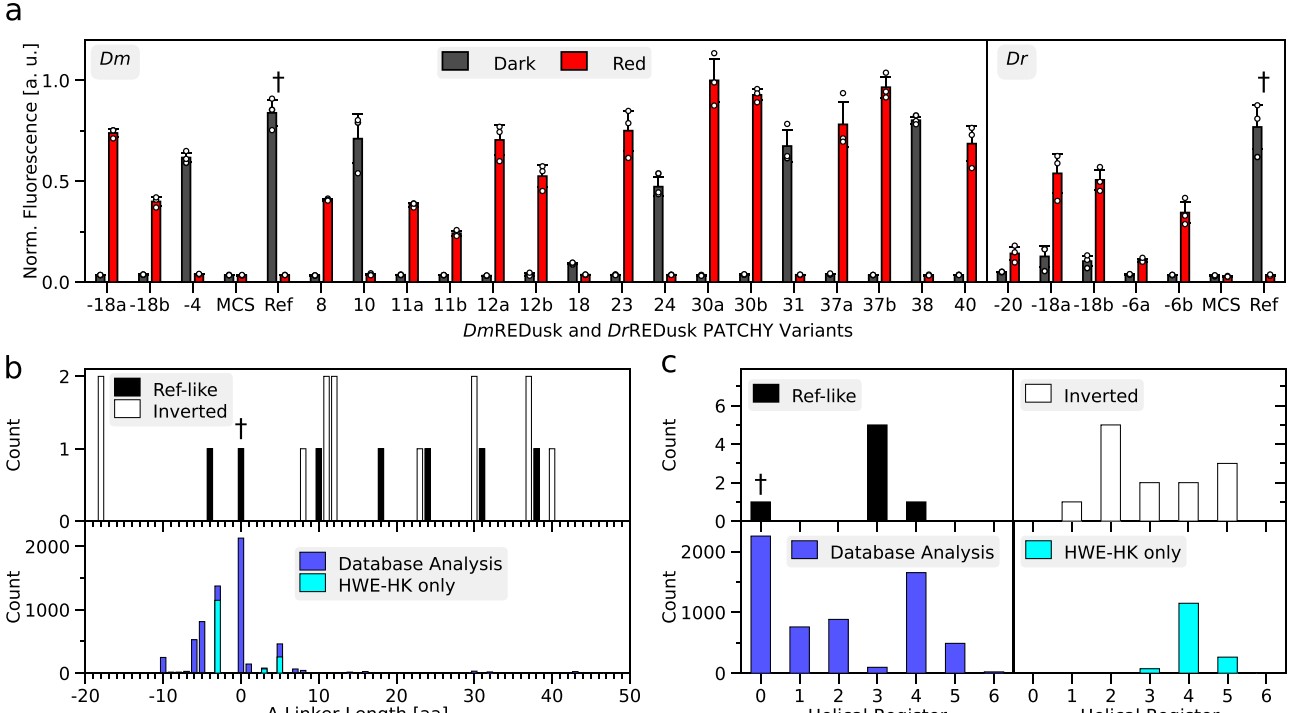

**Fig. 3 | Analysis of *Dm*REDusk and *Dr*REDusk variants with modified linkers.** **a** Bacteria harboring different *Dm*REDusk (left) and *Dr*REDusk (right) systems were incubated under red light (red bars) or in darkness (gray). The *Ds*Red reporter fluorescence was determined as the mean ± s.d. of *n* = 3 biologically independent replicates and compared to that of *Dm*REDusk, *Dr*REDusk (both denoted 'Ref' and labeled with †), and of controls lacking the fluorescent protein but containing a multiple-cloning site (MCS) instead. The relative length of the linker connecting the photosensory module (PSM) and the effector moiety of the SHK variants is indicated on the abscissa. Variants with identical linker lengths but different sequences are designated with the suffixes 'a' and 'b'. The actual linker sequences are provided in Suppl. Fig. 7b. **b** Distribution of relative linker lengths of *Dm*REDusk variants (top panel), with red-light-repressed (termed 'Ref-like') and inverted, red-light-activated specimens (termed 'Inverted') shown as black and white bars, respectively. The dagger symbol (†) denotes the linker length in *Dm*F1 that underlies the original *Dm*REDusk. The bottom panel shows in comparison the linkers in naturally occurring BphP SHKs. Separate analyses consider all histidine-kinase families (blue) or only HWE SHKs (cyan). **c** Analysis of the linker lengths in panel b according to their heptad coiled-coil register. To this end, the distribution is plotted against the remainder after division by seven of the linker length (that is, the modulo 7). Color codes as in panel **b**. Source data are provided as a Source Data file.

of TAMRA fluorescence anisotropy. In darkness, *Dm*F1 and *Dr*F1 prompted anisotropy increases of 0.08 h⁻¹ and 0.11 h⁻¹, respectively, upon ATP addition, indicative of FixJ phosphorylation (Fig. 2f, Suppl. Fig. 6). Consistent with the reporter-gene experiments and the Phos-tag assays, these data reflect that both receptors act as net kinases in darkness (i.e., in the Pr state). Analogous to YF1[23,39], exposure to red light switched the activity to that of a net phosphatase, and the anisotropy signal dropped. Intriguingly, the phosphatase activity was markedly stronger in *Dm*F1 with an anisotropy decrease at a rate of −0.30 h⁻¹ compared to *Dr*F1 with a rate of −0.06 h⁻¹ which immediately accounts for the different light sensitivities of *Dm*REDusk and *Dr*REDusk. That is because the phosphatase activity of *Dm*F1 under red light (in its Pfr state) greatly outweighs its kinase activity in darkness (in its Pr state). Consequently, only a relatively small fraction of *Dm*F1 needs to be converted to Pfr to tilt the kinase-phosphatase balance to net phosphatase at the ensemble level. Ergo, less red light is required for actuation of the *Dm*REDusk circuit than for *Dr*REDusk (see Fig. 1c).

## Linker modifications as a means of reprogramming two-component output

The above experiments showcase that modulation of the kinase-phosphatase duality can greatly impact on SHK activity and TCS output. Spurred by these findings, we explored modifications of the linkers connecting the PSM and DHp/CA moieties of the SHKs. As noted above, said linkers commonly adopt α-helical coiled-coil structures within the homodimeric SHK, which was recently confirmed for the *Dr*BphP by cryo-electron microscopy (Suppl. Fig. 1)[20].

These linkers serve as crucial conduits in signal transduction, and alterations of their length and sequence suffice to utterly alter the kinase-phosphatase equilibrium and thereby reprogram TCS signal output[27,40,41]. We thus applied the PATCHY method[27] to generate a library of *Dm*F1 variants with their linkers extended or shortened by up to 40 residues. As detailed before[27], PATCHY uses PCR amplification with sets of staggered forward and reverse primers to produce SHK variants differing in the length and sequence of their linkers connecting the BphP PSM and histidine-kinase effector moieties (Suppl. Fig. 7a). Importantly, the linker sequences are not varied at random but derive from the linkers of the parental receptors, that is, *Dm*BphP and *B. japonicum* FixL in the case of *Dm*F1. PATCHY library generation was conducted within the *Dm*REDusk context, thus enabling the ready screening under dark and red-light conditions for light-responsive variants. Doing so identified six unique variants that exhibited light responses similar to *Dm*REDusk, i.e., greater reporter-gene expression in darkness than under red light (Fig. 3a, Suppl. Fig. 7b). We refer to these variants by their linkers lengths relative to *Dm*F1. For instance, the variant *Dm*F1-4 has a linker four residues shorter than that of *Dm*F1. Pairs of certain variants had the same length of the linker but different sequences, which we designate by the suffixes 'a' and 'b', e.g., in the case of the variants *Dm*F1-18a and *Dm*F1-18b (Fig. 3a, Suppl. Fig. 7b). Most variants exhibiting light-repressed reporter expression possess linkers longer than that in *Dm*F1 by up to 38 residues (Fig. 3b). Given the coiled-coil nature of the linker with two α-helical turns per seven residues, we also analyzed the linker lengths in terms of their heptad periodicity (Fig. 3c)[27].

Strikingly, most variants populated a heptad register shifted by +3 residues relative to *Dm*F1. That is, their linkers were extended or shortened by $\pm 7 \times n + 3$ residues compared to *Dm*F1, where $n$ is an integer number.

Intriguingly, the PATCHY library also contained 13 unique *Dm*REDusk variants with an inverted response to red light, i.e., they prompted higher reporter expression under red light than in darkness (Fig. 3a, Suppl. Fig. 7b). Whereas two of the inverted variants had linkers that are 18 residues shorter than in *Dm*F1, the remainder exhibited more extended linkers. In contrast to the above light-repressed variants, the linker lengths of the inverted, light-activated variants did not concentrate on a single heptad register but spanned several. We extended our analyses to the *Dr*REDusk context and, by again applying PATCHY, identified five unique variants affording red-light-activated gene expression (Fig. 3a, Suppl. Fig. 7b). All these variants possessed shorter linkers than the parental *Dr*F1 and fell into the linker registers +1 and +3. The broad distribution of linker registers in the light-activated *Dm*F1 and *Dr*F1 variants contrasts with that observed for YF1 where signal-inverted, light-activated variants essentially fell into the helical register +1 relative to the original, light-repressed SHK[23,27].

We wondered whether the predominance of certain linker lengths and registers among the *Dm*F1 and *Dr*F1 variants identified by PATCHY correlates with the lengths of such linkers in naturally occurring BphP-coupled SHKs. A database analysis (Fig. 3b) revealed that the lengths of these linkers clustered within around ±10 residues relative to the parent *Dm*F1 and *Dr*F1 receptors. The linkers populated several heptad registers, merely the +3 and +6 registers were notably under-represented (Fig. 3c). Interestingly, the +3 linker register into which the majority of light-repressed *Dm*F1 variants fall is thus but poorly represented among the natural BphP-coupled SHKs. The principal analysis results were similar when excluding HWE histidine kinases that may employ different signal-transduction mechanisms[42,43]. Taken together, the linkers presently identified by PATCHY therefore cover regimes that are only infrequently, if at all, used in nature. Whereas the underlying reasons for the comparatively narrow linker length distribution in the natural receptors remain unclear, at the very least our data indicate that linkers outside the natural parameter space can principally support intact SHK signal transduction as well.

**Inverted two-component systems facilitate optogenetics and biotechnology**

Inversion of the response to light is not only mechanistically interesting, but also it provides opportunities for applications in biotechnology and optogenetics[32]. Based on particularly stringent regulatory responses, we subsequently focused on *Dm*F1 and *Dr*F1 variants with linkers 23 residues longer and 6 residues shorter, respectively, than their parental systems (denoted as '23' and '−6b' in Fig. 3a and Suppl. Fig. 7b), and we refer to these variants as *Dm*F1+23 and *Dr*F1-6b. Given their inverted light responses, the associated optogenetic circuits are designated as *Dm*DERusk and *Dr*DERusk in the following. Both circuits afford qualitatively similar responses as *Dm*REDawn and *Dr*REDawn, i.e., induction of expression by red light (Suppl. Fig. 3), but do away with the λ phage cI repressor. Removal of this repressor offers at least two advantages: first, it reduces the metabolic burden on the bacteria; second, it facilitates the combination with genetic circuits that rely on this repressor. Given these benefits, we characterized in detail the response to red light of bacteria harboring the *Dm*DERusk and *Dr*DERusk systems (Fig. 4a, b). In case of *Dm*DERusk, the *Ds*Red reporter fluorescence was low in darkness but monotonically increased with red light by 300-fold and with a half-maximal light dose of $(2.0 \pm 0.1) \, \mu W \, cm^{-2}$. For *Dr*DERusk, the reporter fluorescence increased around 90-fold under red light with a half-maximal dose of $(0.60 \pm 0.04) \, \mu W \, cm^{-2}$ (Fig. 4a). Perplexingly, the *Dr*DERusk circuit based on the *Dr*PSM thus afforded higher light sensitivity than *Dm*DERusk based on the *Dm*PSM, which contrasts

with the observations made for *Dr*REDusk vs. *Dm*REDusk (see Fig. 1c). Flow cytometry revealed uniformly low reporter fluorescence in darkness for *Dm*DERusk and *Dr*DERusk that was 1.5-fold and 1.8-fold, respectively, above the background level determined for an empty-vector control (MCS) (Fig. 4b). Red light induced a uniform shift of the single-cell fluorescence distribution to 70-fold and 51-fold higher values, respectively. Thus, both systems have low basal expression (i.e., leakiness), and support stringent regulation by red light in an essentially all-or-none manner.

To unravel the molecular basis for the unexpected higher light sensitivity in *Dr*DERusk compared to *Dm*DERusk, we expressed and purified the underlying SHKs *Dr*F1-6b and *Dm*F1+23. Absorbance spectroscopy confirmed BV chromophore incorporation and light-driven Pr⇌Pfr interconversion as in the parental *Dr*F1 and *Dm*F1 SHKs (Suppl. Fig. 8). Notably, *Dr*F1-6b and *Dm*F1+23 incorporated BV to comparable extents and underwent Pr→Pfr conversion under red light with closely similar speed and efficiency (Suppl. Fig. 8). We next assessed the enzymatic activity of these SHKs in darkness and under red light by Phos-tag analysis (Fig. 4c). Consistent with their inverted response within the reporter-gene context (see Fig. 4a), both *Dr*F1-6b and *Dm*F1+23 showed responses to light opposite those seen for *Dr*F1 and *Dm*F1 (see Fig. 2e) in that substantial FixJ phosphorylation occurred under red light but not in darkness. To obtain quantitative insight, we assessed the enzymatic activity and light response of these SHKs by fluorescence anisotropy (Fig. 4d, Suppl. Fig. 9). Under red light, both *Dr*F1-6b and *Dm*F1+23 exhibited net kinase activity with anisotropy increases of around $0.14 \, h^{-1}$ and $0.18 \, h^{-1}$, respectively. Exposure to far-red light converted both variants to net phosphatases. Notably, the phosphatase activity of *Dm*F1+23 (fluorescence anisotropy decrease of $-0.12 \, h^{-1}$) trumped that of *Dr*F1-6b ($-0.02 \, h^{-1}$) which can account for the differing light sensitivities of *Dm*DERusk compared to *Dr*DERusk. Given the comparatively weak phosphatase activity of *Dr*F1-6b, red-light-induced Pr→Pfr conversion of a relatively small SHK portion suffices for net kinase activity at the ensemble level. In the case of *Dm*F1+23, the phosphatase activity in the dark-adapted state is stronger, and consequently a larger SHK fraction needs to be converted to the Pfr state to result in net kinase activity.

We next addressed whether these systems support multiplexing with optogenetic circuits that drive gene expression in response to other light colors. We previously combined *Dr*REDusk and *Dr*REDawn with the blue-light-responsive pCrepusculo and pAurora circuits[28,44,45] (Suppl. Fig. 10a). Briefly, the latter two setups rely on the light-oxygen-voltage receptor PAL[46] that sequence-specifically binds a small RNA molecule, termed aptamer, once triggered by blue light. In pCrespusculo[45], this aptamer is interwoven with the ribosome-binding site of a target gene, and thereby PAL represses the translation and expression of this gene under blue light. pAurora inverts and amplifies the response to blue light by including a gene-inversion cassette based on the λ phage cI repressor (Suppl. Fig. 10a). Given that pAurora and *Dr*REDawn share the cI component, crosstalk is fully expected, irrespective of these circuits responding to different light colors. To demonstrate this aspect, we combined in the same bacterial cells a pAurora plasmid driving *Ds*Red expression and a *Dr*REDawn plasmid controlling the expression of the yellow-fluorescent YPet protein (Suppl. Fig. 10b, c). Said bacteria were exposed to combinations of blue and red light, and the *Ds*Red and YPet fluorescence was measured. Compared to darkness, both fluorescence readings monotonically increased with the doses of blue and red light. Owing to the crosstalk, the *Ds*Red and YPet fluorescence rose in lockstep, and the maximum values were only reached in the presence of both blue and red light. While potentially of utility in some applications, commonly crosstalk is undesired. When combining pAurora-*Ds*Red with *Dm*DERusk-YPet (Fig. 5a), the crosstalk was eliminated, and the circuits could be induced by blue and red light, respectively, largely independently of each other (Fig. 5b, c).

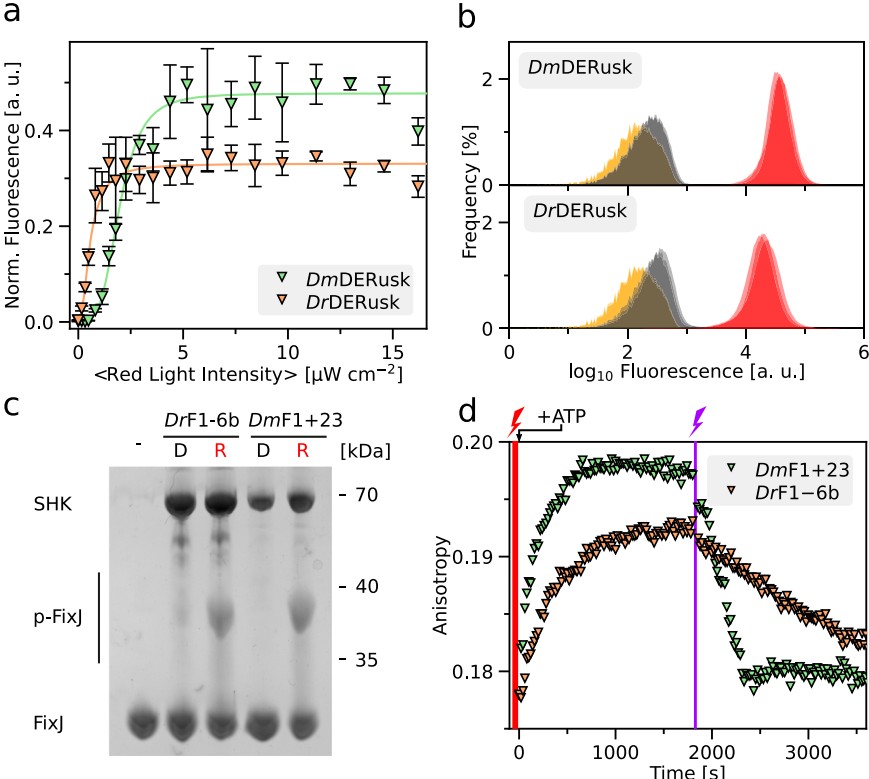

**Fig. 4 | Analysis of the inverted circuits *Dm*DERusk and *Dr*DERusk. a** *Ds*Red production of bacteria harboring *Dm*DERusk (light green triangles) or *Dr*DERusk (light orange triangles) as a function of red-light intensity. Fluorescence readings were normalized to the optical density of the bacterial cultures at 600 nm ($OD_{600}$) and corrected for background fluorescence. Data represent mean ± s.d. of $n = 3$ biologically independent replicates and were normalized to *Dm*REDusk in darkness. Light intensities are averaged over the duty cycle as marked by angled brackets. **b** Analysis by flow cytometry of bacteria carrying *Dm*DERusk (top) or *Dr*DERusk (bottom) under red light (red) or in darkness (gray). The yellow curves denote control bacteria harboring empty vectors with a multiple-cloning site replacing *Ds*Red. $n = 3$ biologically independent replicates for each sample are displayed. **c** The sensor histidine kinases (SHK) *Dm*F1+23 and *Dr*F1-6b,

underpinning *Dm*DERusk and *Dr*DERusk, respectively, were analyzed on Phos-tag gels as described in Fig. 2e. Both SHKs promote the phosphorylation of the response regulator FixJ under red light compared to darkness, consistent with the reporter-gene data shown in panels a and b. **d** The enzymatic activity of the *Dm*F1+23 and *Dr*F1-6b SHKs was also monitored by fluorescence anisotropy (see Fig. 2f, Suppl. Fig. 9a). Illuminated with red light (indicated by red flash), both variants exhibit net kinase activity and phosphorylate FixJ, as revealed by rising TAMRA fluorescence anisotropy. Exposure to far-red light (purple flash) converts both *Dm*F1+23 and *Dr*F1-6b to net phosphatases that promote the dephosphorylation of FixJ, as reflected in decreasing fluorescence anisotropy. Experiments were done in triplicate ($n = 3$) with similar outcomes (see Suppl. Fig. 9b, c). Source data are provided as a Source Data file.

## Discussion

### Two-component systems with altered light response for biotechnology

The recent years have witnessed the advent of innovative applications in biotechnology, synthetic biology, materials science, and even theranostics that bank on light-responsive bacteria[32]. Among the optogenetic modalities available in bacteria, the light-dependent control of gene expression appears most versatile and widespread. Optogenetic circuits responding to red light, including pREDusk, pREDawn[28], and the systems developed at present, afford at least two principal benefits. First, they lend themselves to multiplexing with a slew of established optogenetic circuits that predominantly respond to blue but not red light (see, e.g., Fig. 5)[32]. Second and arguably even more important, red light penetrates living tissue to much larger extent than visible light of shorter wavelengths[47]. This aspect comes to bear in scenarios where light delivery in situ is limiting, for instance in theranostic applications but also in material science. As a case in point, we recently demonstrated that pREDawn reacts to light very sensitively and can hence be actuated by red light through tissue mimics with the optical properties of skin or skull at light intensities around 200-fold below the therapeutically allowed safe limit[28]. By exchanging the original PSM of the *Dr*BphP for that of the *Dm*BphP, we boosted the light sensitivity in the resultant *Dm*REDusk and *Dm*REDawn systems by one order of

magnitude. Put another way, tenfold less light is required to activate these circuits to the same extent as the corresponding setups based on the *Dr*BphP PSM.

Arguably, the majority of optogenetic applications demand light-induced activation of downstream responses, such as gene expression, rather than inactivation. Via introduction of a gene-inversion cassette based on the λ phage cI repressor[34,36,48], we hence inverted and concomitantly amplified the expression output of the *Dm*REDusk circuit. In marked contrast to the related blue-light-sensitive YF1/FixJ TCS[36], but in common with the earlier *Dr*REDawn system[28], doing so elevated the basal output and thereby reduced the dynamic range of light regulation (i.e., the ratio of expression in darkness and light). Moreover, the analysis of single-cell fluorescence revealed partial leakiness of the *Dr*REDawn and *Dm*REDawn setups in darkness, which, depending on use case, may severely curtail the desired light-induced effect. While the underlying molecular reasons are not known in full detail, the comparatively fair performance may owe to the reliance on the strong cI repressor which is constantly expressed under dark conditions.

Through systematic variation of the linker interconnecting the PSM and HK moieties in the *Dm*F1 and *Dr*F1 histidine kinases, we pinpointed inverted receptor variants that prompt elevated gene expression under red light compared to darkness. Of particular

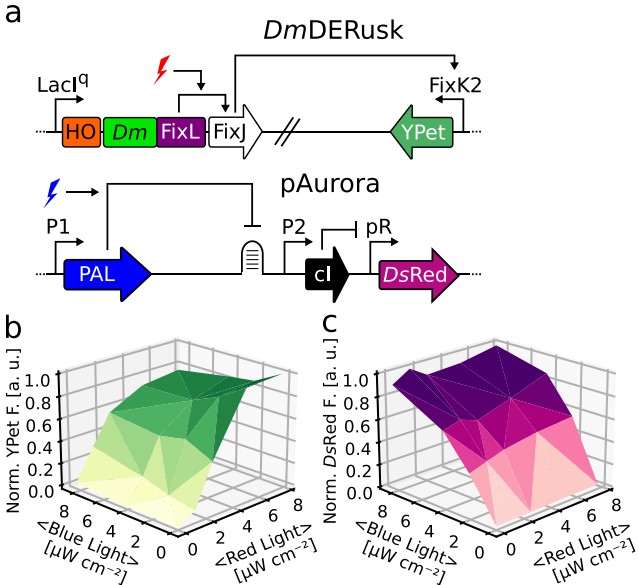

**Fig. 5 | Multimodal control of bacterial gene expression by red and blue light.** **a** Schematics of the red-light-responsive *Dm*DERusk circuit, developed at present, and of the earlier pAurora system that reacts to blue light[45]. **b, c** Bacteria harboring both pAurora-*Ds*Red and *Dm*DERusk-YPet were incubated under different red and blue light intensities. The YPet (**b**, green) and *Ds*Red (**c**, pink) fluorescence readings were normalized to the optical density of the bacterial cultures at 600 nm ($OD_{600}$) and corrected for background fluorescence. Data represent the mean of $n = 3$ biologically independent replicates. Light intensities are averaged over the duty cycle as marked by angled brackets. Source data are provided as a Source Data file.

advantage, the *Dm*DERusk and *Dr*DERusk setups elicit responses to light opposite those triggered by the parental setups they derive from; they maintain high dynamic range of light regulation of up to several hundredfold; they offer simpler architecture with a smaller genetic footprint than *Dm*REDawn and *Dr*REDawn; they operate without any gene-inversion cassettes (thus alleviating the impact on the host bacterium); and therefore they are less prone to cross-talk with other genetic circuits which are frequently reliant on the near-ubiquitous cI phage repressor. A recent survey of approaches for regulating by light bacterial gene expression[32] indicates that the *Dm*DERusk and *Dr*DERusk systems substantially and timely add to the optogenetic repertoire. That is because existing setups responding to red light are relatively scarce in the first place; they frequently lower target gene expression under light rather than increasing it[49–51]; and they often employ the reduced bilin phycocyanobilin (PCB), formed out of biliverdin via an additional enzyme[48,50–52]. *Dm*DERusk and *Dr*DERusk outperform several red-light-responsive gene-expression circuits but they fall slightly short of the CcaRS TCS which harnesses a cyanobacteriochrome receptor with PCB chromophore to react to red and green light[48,51,52]. That notwithstanding, numerous applications of the CcaRS TCS, pDawn[36], and other circuits[32] suggest considerable scope for the red-light-responsive gene-expression circuits engineered at present. Of particular promise, light-responsive bacteria can be administered to the digestive tract of animals to serve as adaptable, living therapeutics[32,53,54]. Given the superior tissue penetration of red compared to blue light[47], the previously required addition of accessory components, such as upconverting nanoparticles[54,55], is obviated and true genetic encoding thereby enabled.

### Remote control and reprogramming of sensor histidine kinases
More broadly, the light-responsive SHKs investigated here serve as experimentally tractable paradigms for the study of allosteric

mechanisms and signaling strategies in natural and engineered TCSs alike. This is exemplified by the *Dm*REDusk and *Dr*REDusk pair which differ tenfold in their light sensitivity (see Fig. 1c), despite the underlying *Dm*F1 and *Dr*F1 SHKs absorbing red light with similar efficiency and undergoing comparable photocycles (see Fig. 2a–d). The investigation of the enzymatic activities of both SHKs in the absence and presence of their signal, i.e., prior to and upon exposure to red light, uncovered the mechanistic principles at play (see Fig. 2e, f). In darkness, both *Dr*F1 and *Dm*F1 dwelled in their K state and acted as net kinases on their response regulator with similar activity levels. Red light incurred a transition to the P state and converted them to net phosphatases. Notably, both receptors not only resort to sensor modules with closely similar traits, but also, they use the exact same histidine-kinase effector moiety. These commonalities notwithstanding, the phosphatase reaction under red light is markedly more efficient in *Dm*F1 than in *Dr*F1 (see Fig. 2f), which directly accounts for the divergent light sensitivity observed in the *Dr*REDusk and *Dm*REDusk circuits (see Fig. 1c). Given the pronounced phosphatase activity of *Dm*F1, it suffices to convert a relatively small fraction of this SHK to its P state to trump the kinase activity of the fraction remaining in the K state (Fig. 6a).

In a similar vein, the variation of the linker intervening the photosensory input module and the catalytically active effector module gave rise to the circuits *Dm*DERusk and *Dr*DERusk with responses to red light opposite those of the parental systems. Again, modifications remote from the effector module and its active site evidently sufficed for utterly reprogramming the circuit output. Whereas the original systems exhibit net kinase activity in darkness, in the inverted systems this is the case only after absorption of red light. Probing by PATCHY revealed that light responsiveness predominantly associates with certain linker lengths (Fig. 3). Consistent with our earlier investigation of a blue-light-responsive TCS[27], these preferences likely reflect the continuous coiled-coil structure of the linker, discussed in more detail below. Intriguingly, our present data reveal that circuit variants with red-light-activated net kinase activity populate more coiled-coil registers than variants with red-light-suppressed net kinase activity (see Fig. 3c). This may indicate that the coiled-coil registers supporting phosphatase activity are not as strictly defined as for kinase activity. However, we caution that this observation is based on SHK variants predominantly exhibiting linker lengths that depart from those observed in natural receptors (see Fig. 3b).

Irrespective of these considerations, the *Dr*DERusk circuit strikingly exhibits higher light sensitivity than *Dm*DERusk (see Fig. 4a). This initially perplexing finding can again be rationalized by the net enzymatic activities of the underlying *Dr*F1-6b and *Dm*F1+23 SHKs (see Fig. 4d). As the phosphatase activity of *Dm*F1+23 in the dark-adapted Pr state is much stronger than that of *Dr*F1-6b, a higher proportion of SHK molecules need to be photoconverted to the Pfr state to shift the balance between the opposing elementary SHK activities to the kinase side (Fig. 6b). Consequently, more light is needed to actuate the *Dm*DERusk circuit than *Dr*DERusk (see Fig. 4a). A comparison of *Dm*F1, *Dr*F1, *Dm*F1+23, and *Dr*F1-6b, reveals that the divergent light sensitivities are primarily rooted in the strength of the phosphatase activity (for *Dm*F1 and *Dr*F1 in the red-light-induced Pfr state, for *Dm*F1+23 and *Dr*F1-6b in the dark-adapted Pr state), whereas the kinase activity exhibits similar magnitude across all four SHKs. At least for the SHKs presently investigated, the *Dm*PSM hence supports stronger phosphatase activity than the *Dr*PSM. It is precisely these ingrained differences in phosphatase activity that account for the divergent light sensitivities of *Dr*REDusk vs. *Dm*REDusk and *Dr*DERusk vs. *Dm*DERusk. Consistent with this notion, earlier analyses had pinpointed the phosphatase activity as decisive in governing signal sensitivity of TCS circuits[56].

Despite recent progress, the precise molecular bases underpinning the conformational transition between the K and P states in

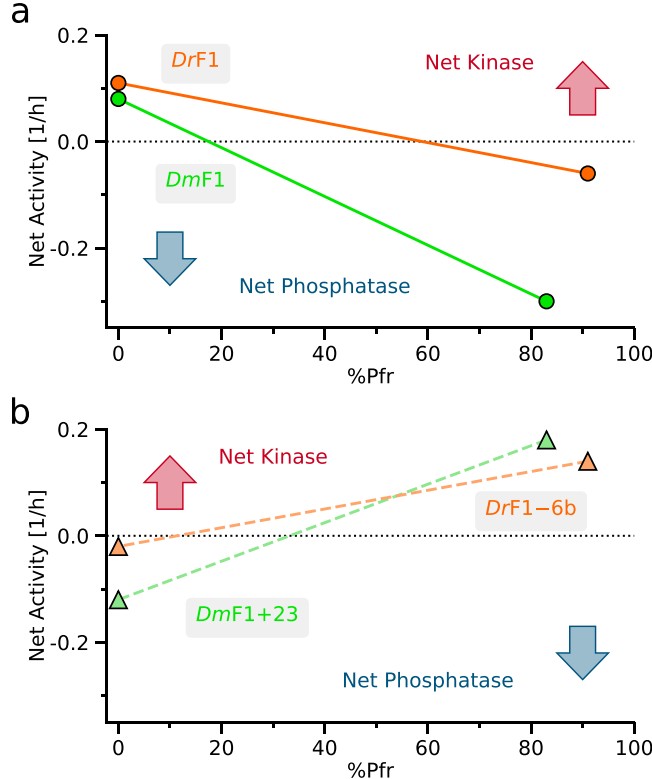

**Fig. 6 | Kinase-phosphatase duality in light-responsive sensor histidine kinases (SHK). a** The net activities of the *Dm*F1 (green circles) and *Dr*F1 SHKs (orange circles) refer to the analyses by fluorescence anisotropy in their dark-adapted K (kinase-active) states and in their partially light-adapted P (phosphatase-active) states following exposure to red light (see Fig. 2f). The fractions of the SHK in the light-adapted Pfr state upon red-light exposure were determined by absorbance spectroscopy (see Fig. 2a, b) and are plotted on the abscissa. Owing to the higher elementary phosphatase activity of the light-adapted P state of *Dm*F1 compared to *Dr*F1, photoconversion of a comparatively small *Dm*F1 fraction already results in net phosphatase activity at the ensemble level. Hence, the *Dm*REDusk circuit is rendered more light-sensitive than *Dr*REDusk (see Fig. 1c). **b** By contrast, the *Dr*DERusk circuit proved more light-sensitive than *Dm*DERusk (see Fig. 4a). These observations are again accounted for by the elementary kinase and phosphatase activities of the underlying *Dr*F1-6b (orange triangles) and *Dm*F1+23 SHKs (green triangles) (see Fig. 4d). As the dark-adapted P state of *Dr*F1 possesses low net phosphatase activity, photoconversion of a relatively small fraction to the light-adapted K state suffices for conversion of the activity to that of a net kinase. Contrarily, *Dm*F1+23 exhibits stronger net phosphatase activity in its dark-adapted P state, and consequently a larger portion needs to be converted to the light-adapted K state to obtain net kinase activity at the ensemble level. Source data are provided as a Source Data file.

bacteriophytochromes, and many related receptors, await full elucidation. The available evidence points at a crucial role of the homodimeric coiled-coil linkers that connect the sensor and effector moieties of SHKs. These linkers provide conduits through which the K/P equilibrium and hence the net activity of the SHK can be controlled at a distance, which is especially relevant for many chemoreceptors that span the plasma membrane. Based on evidence on the thermosensing DesK from *Bacillus subtilis*[19], we have proposed that in several photoreceptor kinases the transition from the K to the P state involves the angular reorientation and altered supercoiling of the α helices making up the coiled coil, which may lead to sequestration of the active-site histidine within the DHp domain[8,40,41]. This principal notion finds recent support in the cryo-EM structures of the *Dr*BphP that resolved light-induced splaying apart of the coiled-coil helices that directly feed into the DHp domain[20]. Residue insertions and deletions within the

linker stand to profoundly alter the coiled-coil geometry, and thereby the basal state of a SHK and its response to signal. While the above mechanistic model accounts for our experimental data, we caution that other models may equally apply. For instance, for different SHKs pivot and piston transitions have been invoked[57].

Independent of the precise nature of the underlying transition between the K and P states, which may well differ across individual receptors, the ability to catalyze both the phosphorylation and dephosphorylation of response regulators appears widely shared across many, if not most, SHKs. Our work underlines the essential role of, and the opportunities afforded by this kinase-phosphatase duality. Perturbations far from the histidine-kinase effector, i.e., within the sensor and the linker domains, can profoundly tilt the K/P equilibrium one way or another. On the one hand, such free-energy perturbations arise from the cognate signals (e.g., light) that a given TCS detects. On the other hand, these perturbations can be introduced at will by SHK modifications far from the active site, thus allowing the rapid exploration of new signal-response regimes. These principles are likely not only relevant for protein engineering in the laboratory, but also, they may have been exploited during evolution. Similar concepts apply to receptors engaged in cyclic-di-GMP second-messenger signaling, too[58–60]. Frequently these receptors possess both GGDEF and EAL entities, which form and break down the second messenger, respectively, and which are commonly regulated antagonistically. Akin to SHKs, highly stringent responses are thereby enabled which may be amenable to the same types of perturbations investigated at present.

## Methods
### Molecular biology
The *Dm*REDusk and *Dm*REDawn plasmids were generated by exchanging the PSM from *D. radiodurans* BphP (Uniprot BPHY_DEIRA, residues 1–506) for the PSM from *D. maricopensis* BphP (Uniprot E8U3T3, residues 1–510) within the previously reported *Dr*REDusk and *Dr*REDawn plasmids[28]. To this end, the gene encoding the *Dm*PSM was amplified by PCR from the template plasmid pQX043 that encodes a light-regulated adenylyl cyclase[30]. The initial *Dm*REDusk and *Dm*REDawn vectors were generated with a *Ds*Red Express2 reporter gene[61] included; plasmid versions with multiple-cloning sites replacing the fluorescent reporter were obtained by PCR amplification and blunt-end ligation.

For protein expression, the genes encoding *Dr*F1 or *Dr*F1-6b were introduced into the pET21b(+) vector (Novagen) between the *Nde*I and *Xho*I restriction sites using the NEBuilder HiFi DNA assembly cloning kit (New England Biolabs). Thereby, an N-terminal T7-tag was removed and a C-terminal hexahistidine (His$_6$) tag installed. For *Dm*F1 and *Dm*F1+23 production, the underlying genes were inserted into the first MCS of the pCDF-Duet plasmid (Novagen) via Gibson assembly[62] and thereby equipped with a C-terminal octahistidine (His$_8$) tag. The second MCS contained the heme oxygenase HO1 from *Synechocystis* sp. responsible for biliverdin production. All sequences were confirmed by Sanger sequencing (Eurofins Genomics, Germany or Microsynth Seqlab, Göttingen, Germany).

### Reporter-gene assays
The response of the various pREDusk, pREDawn, and pDERusk systems to different red-light intensities was measured as before[28]. Briefly, plasmids encoding a given system were transformed into chemically competent *Escherichia coli* CmpX13 cells[63]. 5 mL lysogeny broth (LB) medium containing 50 µg mL$^{-1}$ kanamycin (Kan) were inoculated with these bacteria. In addition to the plasmid variants harboring the *Ds*Red Express2 red-fluorescent reporter (abbreviated as *Ds*Red subsequently)[61], corresponding controls lacking the fluorescence protein and carrying a MCS instead were assessed as well. The cultures were incubated for 24 h at 30 °C and 225 rpm agitation under non-inducing conditions (i.e., 100 µW cm$^{-2}$ 650-nm light for pREDusk

variants; or darkness for pREDawn and pDERusk species). Subsequent handling of the cultures was done under non-inducing light conditions (i.e., red light for pREDusk variants; or green light for pREDawn and pDERusk species). Upon hundredfold dilution of the cultures in 20 mL LB/Kan medium, 200-μL aliquots were dispensed into 64 wells of clear-bottom, black-walled microtiter plates (MTP) (μClear, Greiner BioOne, Frickenhausen, Germany). The MTPs were sealed with a gas-permeable membrane (BF-410400-S, Corning, New York, USA) and illuminated from below with an $8 \times 8$ array of light-emitting diodes (LED) with a peak wavelength of $(624 \pm 8)$ nm for the red emission channel[35,64]. Light intensities were calibrated with a power meter (model 842-PE equipped with a 918D-UV-OD3 silicon detector, Newport, Darmstadt, Germany). All experiments resorted to intermittent illumination with different duty cycles. The $Dm$REDusk, $Dr$REDusk, $Dm$REDawn, $Dr$REDawn, $Dm$DERusk, and $Dr$DERusk circuits were illuminated at duty cycles of 1:10 (i.e., 20 s red light, followed by 180 s darkness) or 1:40 (5 s red light, 195 s darkness). The MTPs were incubated at 37 °C and 750 rpm agitation while being periodically illuminated. After 18 h, the optical density at $(600 \pm 9)$ nm ($OD_{600}$) of the cultures and their $Ds$Red fluorescence were measured with a Tecan Infinite M200 Pro MTP reader (Tecan Group, Ltd., Männedorf, Switzerland). The excitation and emission wavelengths for fluorescence detection were $(554 \pm 9)$ nm and $(591 \pm 20)$ nm, respectively. The fluorescence readings were normalized to $OD_{600}$, averaged, and corrected for the background fluorescence of the corresponding MCS control cultures. Data represent mean ± s.d. of three biologically independent replicates and are normalized to the reporter of fluorescence of $Dm$REDusk in darkness. Data were plotted as a function of light intensity (averaged over the duty cycle) and were fitted to Hill binding isotherms with the Fit-o-mat software[65].

## Flow cytometry

The above *E. coli* CmpX13 cells bearing different pREDusk, pREDawn, and pDERusk systems and their matching MCS controls were grown overnight at 37 °C on LB/Kan agar plates. 200 μL LB/Kan medium were inoculated with single colonies and grown in MTPs for 21–24 h at 30 °C and 750 rpm agitation under non-inducing conditions (i.e., 100 μW cm⁻² 650-nm light for pREDusk variants; or darkness for pREDawn and pDERusk species). After incubation, the cells were diluted 1:100 in LB/Kan medium and split into two cultures for each variant. One culture each remained under non-inducing conditions, while the other culture was transferred to inducing conditions (i.e., darkness for pREDusk variants; 100 μW cm⁻² 650-nm light for pREDusk variants; and 50 μW cm⁻² 630-nm light for pDERusk variants). In the case of $Dr$REDawn and $Dm$REDawn, the cultures were first incubated for 2 h under non-inducing conditions before transfer to the inducing conditions. All cultures were incubated for a total of 18 h at 37 °C and 750 rpm agitation. The cultures were then diluted 20-fold in phosphate-buffered saline (1× sheath fluid, Bio-Rad) and analyzed on a S3e cell sorter (Bio-Rad). The $Ds$Red fluorescence was detected with excitation lasers operating at 488 and 561 nm and an emission channel at $(585 \pm 15)$ nm. For each sample, at least 200,000 events were recorded. The frequency distribution of the decadic logarithm of the single-cell fluorescence data was fitted to skewed Gaussian distributions using Fit-o-mat[65].

## Protein purification

$Dr$F1 and $Dr$F1-6b were expressed and purified as described previously[66]. Briefly, the variants were expressed in *E. coli* strain BL21(DE3) overnight at 22 °C. Cells were then lysed with EmulsiFlex® (Avestin) and incubated overnight on ice with a 10x molar excess of biliverdin hydrochloride (Frontier Scientific) under the exclusion of light. Affinity purification was conducted with HisTrap Fast-Flow Crude columns (GE Healthcare), and size-exclusion chromatography was performed using a HiLoad Superdex 200 pg column (GE Healthcare)

equilibrated with running buffer [30 mM Tris/HCl, pH 8.0]. The proteins were concentrated to approximately 30 mg mL⁻¹ and flash-frozen in liquid nitrogen. To produce $Dm$F1 and $Dm$F1+23, the pCDFDuet-$Dm$F1/HO or pCDFDuet-$Dm$F1+23/HO expression construct was transformed into chemically competent *E. coli* LOBSTR cells[67]. 5 mL LB medium containing 100 μg mL⁻¹ streptomycin (Strep) was inoculated with a single colony and incubated overnight at 37 °C and 40 rpm rotation. 800 mL LB/Strep medium was inoculated with the entire 5-mL culture. The bacteria were grown in baffled Erlenmeyer flasks at 37 °C and 225 rpm agitation. At an $OD_{600}$ of around 0.9, the expression was induced by adding 1 mM isopropyl-β-thiogalactopyranoside (IPTG). To support biliverdin biosynthesis, 0.5 mM $\delta$-aminolevulinic acid were added as well. Expression was performed at 16 °C and 225 rpm agitation for 62 h. The following steps were performed under green safe light. Cells were harvested by centrifugation and resuspended in lysis buffer [50 mM Tris/HCl, 20 mM NaCl, 10 mM imidazole, 1 mM β-mercaptoethanol, pH 8.0, supplemented with protease inhibitor mix (Roche)]. Cell lysis was performed by addition of 1 mg mL⁻¹ lysozyme and sonication. After centrifugation, the cleared lysate was applied to a HiTrap TALON crude column with cobalt-nitrilotriacetic resin (Cytiva), and the His₈-tagged $Dm$F1 was eluted by an imidazole gradient from 0 to 0.5 M. Fractions were analyzed for protein content, purity, and biliverdin incorporation by denaturing SDS polyacrylamide gel electrophoresis in the presence of 1 mM $Zn^{2+}$ [68]. The pooled fractions were dialyzed at 4 °C overnight against SEC buffer [30 mM Tris/HCl, 20 mM NaCl, 1 mM dithiothreitol, pH 8.0]. Following concentration via spin filtration, the sample was applied to a HiPrep 16/60 Sephacryl S-200 HR (GE Healthcare) size-exclusion column. Fractions were analyzed as before, pooled, and dialyzed against storage buffer [20 mM Tris/HCl, 20 mM NaCl, 10 % (w/v) glycerol, 1 mM dithiothreitol, pH 8.0]. Protein was concentrated via spin filtration, flash-frozen in liquid nitrogen, and stored at −80 °C. The protein concentration was determined by UV-vis absorbance spectroscopy using extinction coefficients at 280 nm of 80,900 and 52,200 M⁻¹ cm⁻¹ for $Dr$F1 and $Dm$F1, respectively.

## Spectroscopic characterization

Absorbance spectroscopy was performed in buffer A [20 mM Tris/HCl, 80 mM NaCl, 2.5 mM MgCl₂, pH 8.0] at 25 °C on a Cary 60 UV-vis spectrophotometer (Agilent). Spectra of 4 μM $Dr$F1, $Dm$F1, $Dr$F1-6b, and $Dm$F1+23 were recorded after 180 s exposure to LEDs emitting far-red $[(800 \pm 18)$ nm, 23 mW cm⁻²] and red light $[(650 \pm 10)$ nm, 4.8 mW cm⁻²]. Absorbance data were normalized to the reading at 700 nm following exposure to far-red light.

To track the kinetics of photoactivation, the samples were illuminated with the red and far-red LEDs at intensities of 1.0 mW cm⁻² and 5.1 mW cm⁻², respectively. To ensure reproducible illumination, the LEDs were mounted above the cuvette in a fixed orientation within a custom 3D-printed adapter. The absorbance signal was monitored over time at 750 nm, normalized to the maximum value, and fitted to single-exponential functions. The experiment was conducted in triplicate. With the resultant time constants averaged, the pure Pfr spectra of $Dr$F1, $Dm$F1, $Dr$F1-6b, and $Dm$F1+23 were calculated, as well as the Pr:Pfr ratio at photostationary state under red light[69].

Prior to recording dark-recovery kinetics, the $Dr$F1 and $Dm$F1 samples were illuminated with far-red light. The sample was then illuminated with red light for 60 s, and the absorbance at 280, 700, 750, and 900 nm was recorded every 5 min. The time course of the absorbance readings was fitted to single-exponential functions. Experiments were conducted in triplicate with similar results. To assess the degree of chromophore incorporation, $Dr$F1 and $Dm$F1 were denatured by incubating in 50 mM Tris/HCl, 50 mM NaCl, 6 M guanidinium chloride, pH 8.0 for 45 min. Samples were illuminated with far-red light for 60 s, and spectra were recorded immediately afterwards.

## Histidine-Kinase Assays

Phos-tag experiments were conducted as described for the *Dr*BphP in[10]. 0.31 mg mL$^{-1}$ *Dr*F1, *Dm*F1, *Dr*F1-6b, or *Dm*F1+23 and the response regulator *Bj*FixJ (also at a concentration of 0.31 mg mL$^{-1}$) were mixed and incubated in 25 mM Tris/HCl pH 7.8, 5 mM MgCl$_2$, 4 mM 2-mercaptoethanol, 5% (v/v) ethylene glycol[10]. The samples were illuminated with a 657-nm red LED (Mightex) for 5 min or a 782-nm far-red laser (Roithner) for 20 s, and the reactions were initiated by adding 1 mM ATP. Samples were incubated in darkness or under red light (657 nm) for 20 min, at which point 5x SDS-PAGE loading buffer containing 1 mM ZnCl$_2$ was added to stop the reaction. Samples were run on a 9% (w/v) denaturing polyacrylamide gel containing 50 μM Phos-tag acrylamide (Wako Chemicals). After the run, the gels were stained with PageBlue (Thermo Scientific). An uncropped version of the Phos-tag gels in Figs. 2e and 4c is supplied as Suppl. Fig. 11, which also includes corresponding results from replicate experiments.

For fluorescence anisotropy measurements, a double-stranded DNA (dsDNA) substrate containing a binding site for phospho-FixJ was prepared by annealing a forward and a complementary reverse primer (5′-GAGCGATATCTTAAGGGGGGTGCCTTACGTAGAACCC-3′). The forward primer was labeled with (5-and-6)-carboxytetramethylrhodamine (TAMRA) at its 5′ end as previously described[70], and the underlined part of the primer sequence denotes the phospho-FixJ binding site[39]. For the enzymatic assay, 2.52 μM *Dr*F1, *Dm*F1, *Dr*F1-6b, or *Dm*F1+23 were mixed with 25.2 μM *Bj*FixJ and 1.26 μM TAMRA-dsDNA in 20 mM Tris/HCl, 80 mM NaCl, 2.5 mM MgCl$_2$, pH 8.0. The reaction mixture was transferred to a black 96-well MTP (FluoroNunc) and equilibrated at 25 °C for 5 min. The sample was then illuminated for 60 s to populate the kinase-active K state (i.e., 800-nm light for *Dr*F1 and *Dm*F1, and 650-nm light for *Dr*F1-6b and *Dm*F1+23) and the kinase reaction was initiated by addition of 1 mM ATP. The anisotropy of the TAMRA fluorescence was monitored with a MTP reader (CLARIOstar, BMG Labtech) at (540 ± 20) nm excitation and (590 ± 20) nm emission for 30 min. Then, the MTP was ejected, and the samples were illuminated for 20 s with a 650-nm LED in case of *Dr*F1 and *Dm*F1, or for 30 s with 800-nm light in case of *Dr*F1-6b and *Dm*F1+23, before continuing the fluorescence measurement for another 30 min.

## Generation and screening of linker libraries

Libraries of *Dm*REDusk and *Dr*REDusk plasmid variants with differing linkers between the PSM and effector modules of their SHKs were generated via the PATCHY protocol[27] (Suppl. Fig. 7a). To this end, starting constructs, denoted *Dm*REDusk_sc and *Dr*REDusk_sc, were prepared with bipartite linkers that were extended relative to *Dm*F1 and *Dr*F1. In case of *Dm*REDusk_sc, this construct comprised residues 1-540 of the *Dm*BphP (Uniprot E8U3T3) connected to residues 257-504 of *B. japonicum* FixL (FIXL_BRAJA); for *Dr*REDusk_sc, residues 1-529 of the *Dr*BphP (BPHY_DEIRA) were used instead. In both starting constructs, a short nucleotide stretch containing a *Ksp*AI restriction site and a one-nucleotide frame shift, was inserted at the junction between the PSM and the effector moieties. Libraries of plasmid variants with serially shortened or elongated linkers were prepared by PCR amplification with staggered primers. A set of forward and reverse primers were generated using a custom Python script (https://github.com/vrylr/PATCHY). The primers for a given construct were pooled and added to the PCR reaction at a final concentration of 0.25 μM. The PCR product was purified by gel extraction and digested with the *Ksp*AI enzyme. The reaction product was purified, phosphorylated with T4 polynucleotide kinase (incubation at 37 °C for 30 min), and ligated with T4 DNA ligase in the presence of PEG 4000 and ATP for 60 min at 25 °C. Following transformation of the library into DH10b cells, the bacteria were plated on LB/Kan and incubated overnight at 37 °C in darkness or under red light (100 μW cm$^{-2}$, 650 nm). Next, the plates were screened for red colonies, indicative of *Ds*Red production. To identify pREDusk variants with inverted light response, colonies that produced *Ds*Red only under red light were selected for DNA sequencing and subsequent analyses.

Individual clones from the PATCHY library were grown in 200 μL LB/Kan within a clear MTP (Nunc, ThermoFisher) for 24 h at 30 °C and 750 rpm agitation. Cultures were then diluted hundred-fold in LB/Kan and split into two separate cultures each, which were then incubated for 18 h at 37 °C and 750 rpm agitation in darkness or under red light (100 μW cm$^{-2}$, 650 nm), respectively. After incubation, the $OD_{600}$ and *Ds*Red fluorescence were measured and evaluated as described above.

## Multiplexing of optogenetic circuits

These experiments were conducted similarly as before[28]. The gene encoding the yellow-fluorescent YPet protein[71] was introduced into the *Dm*DERusk plasmid by Gibson cloning[62]. Next, *E. coli* CmpX13 bacteria were co-transformed with pAurora-*Ds*Red[45] in combination with either *Dr*REDawn-YPet[28] or *Dm*DERusk-YPet (this work). Notably, in the pAurora plasmid, *Ds*Red expression is placed under the control of the LOV receptor PAL and activates under blue light. Bacterial starter cultures harboring combinations of these plasmids were prepared in LB medium supplemented with 50 μg mL$^{-1}$ kanamycin and 100 μg mL$^{-1}$ streptomycin (LB/Kan+Strep). Following 24 h incubation in darkness at 30 °C and 225 rpm agitation, the cultures were diluted 100-fold into fresh LB/Kan+Strep medium. 200 μL each of the diluted culture were dispensed into 64 wells of a μClear MTP which was sealed as described above and then placed on top of a programmable matrix of light-emitting diodes, see above[35]. For these experiments, the matrix was configured such that it delivered from below blue light [(463 ± 12) nm] and/or red light [(624 ± 8) nm] to individual wells of the MTP. Both light colors were applied intermittently at a 1:10 duty cycle, i.e., phases of 20 s illumination, followed by 180 s darkness. After 18 h incubation at 37 °C and 750 rpm agitation, the optical density at 600 nm of the cultures and their *Ds*Red fluorescence were measured as described above. The YPet fluorescence was determined likewise but using excitation and emission wavelengths of (500 ± 9) nm and (554 ± 9) nm, respectively. The fluorescence readings were normalized by the optical density and averaged over three biologically independent replicates. The YPet and *Ds*Red reporter fluorescence readings were plotted as functions of the blue-light and red-light intensities (averaged over the duty cycle) using custom Python scripts[28].

## Database analyses

We based the linker-length analyses of histidine kinases coupled to a PHY domain (phytochrome-specific) on the HMM profiles for the Pfam families[72,73] PHY (PF00360), HisKA (PF00512), HisKA_2 (PF07568), HisKA_3 (PF07730), HWE_HK (PF07536), and H-kinase_dim (PF02895). Using HMMer[74], we parsed the Pfam clan PF00360 and extracted 6,164 occurrences of a PHY domain within 100 residues upstream of either a HisKA, HisKA_2, HisKA_3, HWE_HK, or H-kinase_dim domain. We also conducted a separate analysis for proteins containing a PHY domain within 100 residues upstream of an HWE_HK domain. The sequences of the linkers intervening the domains were then aligned using MUSCLE[75]. Using custom Python scripts, we determined the lengths of the aligned linkers by counting the number of residues between two reference positions, corresponding to residues W483 (within the PHY domain, see Suppl. Fig. 2) and H532 (within the DHp domain) of the *Dr*BphP. Histograms of the absolute linker length and the remainder after division by 7 (i.e., modulo 7) were plotted with Fit-o-mat[65].

## Statistics and reproducibility

For all measurements, at least three biological replicates (i.e., independent samples) were analyzed and averaged. No statistical tests were used to determine sample size. The sample sizes are stated for each experiment. They were determined based on the size and distribution of the measurable differences in the experimental observable. No data were excluded from the analyses. All experiments were

**Article** https://doi.org/10.1038/s41467-024-49251-8

repeated at least *n* = 3 times with similar outcome. Details are provided in the figure legends.

## Reporting summary

Further information on research design is available in the Nature Portfolio Reporting Summary linked to this article.

## Data availability

The experimental data generated in this study have been deposited in the Zenodo repository under accession code 10680511. These data are also provided in the Source Data file. The structural data used in Suppl. Fig. 1 of this study are available in the Protein Data Bank under accession codes 8AVW and 2C2A. The protein sequences underlying the alignment shown in Suppl. Fig. 2 are available from the Uniprot database under accession numbers E8U3T3 and Q9RZA4. The *Dm*REDusk, *Dm*DERusk, and *Dr*DERusk plasmids and their sequences have been deposited with Addgene (https://www.addgene.org) under the accession numbers 213701, 213702, and 213703, respectively. Source data are provided with this paper.

## Code availability

Python scripts are available for the analyses of linker properties within a multiple sequence alignment (https://github.com/TheAngulion/eval_msa)[76] and for the evaluation and presentation of data from two-color reporter gene experiments (https://github.com/TheAngulion/multiplex)[77]. Scripts to design PATCHY primers are available at: https://github.com/vrylr/PATCHY. The Fit-o-mat software can be obtained from: https://github.com/TheAngulion/fit-o-mat.

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

## Acknowledgements

We thank Dr. Q. Xu for cloning *Dm*REDusk. We greatly appreciate financial support by the European Commission (FET Open NEUROPA, grant 863214 to A.M.), the Deutsche Forschungsgemeinschaft (grant MO2192/4-2 to A.M.), the Research Council of Finland (grant 330678 to H.T.), and the Finnish Cultural Foundation (grant 00220697 to E.M.). Publication fees partially funded by the Open Access Publishing Fund of the University of Bayreuth.

## Author contributions

S.S.M.M. designed, conducted, and evaluated experiments, prepared figures, and drafted and edited the manuscript; E.M. designed, conducted, and evaluated experiments, prepared figures, and drafted and edited the manuscript; A.R. designed, conducted, and evaluated experiments; H.T. acquired funding, supervised the study, and edited the manuscript; A.M. acquired funding, supervised the study, designed, conducted, and evaluated experiments, and drafted and edited the manuscript.

## Funding

## Competing interests

The authors declare no competing interests.
