## [Peer Review File · Nature Communications]

REVIEWER COMMENTS

Reviewer #1 (Remarks to the Author):

In this manuscript the authors increase the sensitivity of their recent red light responsive bacterial gene expression systems pREDawn (Red OFF) and pREDDusk (Red ON) by swapping the photosensory domain of the *Deinococcus radiodurans* bacteriophytochrome-derived photoreceptor BphP from which these systems are built with that of *Deinococcus maricopensis*, which had been identified to have robust signaling properties in a previous screen. They go on to characterize the photoresponse and phosphorylation signaling in bacteria and in vitro and convincingly show that the increase in sensitivity is due to altered signal transduction between the photosensor and phosphoryl-signaling domains. In doing so, they build and characterize a number of other variants of the engineered *D. maricopensis* photoreceptor kinase with different signaling and thus gene regulatory properties. In particular, they construct libraries of varied linker lengths and begin to identify principles of linker periodicity and register that play a role in kinase signaling logic. I agree with the authors that the increased light sensitivity of their new variants is useful for optogenetics research, particularly when light intensity is limiting (e.g. in vivo, potentially in a bioreactor). Overall, the paper is a valuable contribution to both the optogenetics and histidine kinase signaling literature.

I do not understand the argument that the authors make about quantum yield not explaining the increase in sensitivity of DmRedDusk versus DrRedDusk (lines 122-126). It seems they are assuming that the Dm PSM and the DrPSM both exhibit quantum yields of 0.1 – 0.2, because that is generally the quantum yield exhibited by BphP PSMs. However, it is not clear there is a priori evidence confirming that both of these PSMs have quantum yields in this range. What if DrPSM had an unusually low quantum yield for example (e.g. 0.01)? That then may explain the difference in light sensitivity. The authors should provide more clearly justify how they can exclude this, or similar, possibilities.

What is the explanation for the bimodality in the dark state for the pDawn systems? It is presumably due to the *ci* inverter circuit as it is absent in the pDusk systems. The higher output gene expression mode of the dark state overlaps with the red light activated DmRedDawn system, which reduces the utility of the system for optogenetics to a degree.

On lines 300-301, I believe the authors are referring to DmDERusk and DrDERusk rather than DmREDusk and DrREDusk.

The authors should explain how the PATCHY library construction method works and the composition of the libraries created by this method as it relates to SHK engineering in the main text. Currently this is not described, making it difficult to understand why the library that was generated was generated.

The SHK nomenclature in Figure 3A is not defined, making the figure difficult to interpret. The authors should define the nomenclature in the main text and/or figure legend. What is the difference between a linker variant named by a number (e.g. 8) versus a number and the letter “a” (e.g. -18a) versus a number and the letter “b” (e.g. -18b)?

I don't think that Figure 5 a – b adds much to the impact of the of the paper. In particular, the authors do not achieve independent control of the expression of two genes with blue and red light using these constructs, due to the fact that they cross-talk through *cl* repressor. I suppose that one could use this system as a blue OR red light gate, but the authors do not comment on that. I recommend they be moved to the supplementary information and replaced with schematics of the engineered genetic devices used in panels c and d.

Reviewer #2 (Remarks to the Author):

Report to 486044_0 443508

The authors target one of the cornerstones in the regulation of microbial lifestyle. Histidine kinases (HKs)-response regulators (RRs), or, TCS (two component systems) are ubiquitous and (i) sense a multitude of external stimuli from the environment, and (ii) give a wide variation of intracellular signals allowing for a physiological response of the living organism. The components of these systems-often still coined as Che- (CheY, -W, -A etc.) from their initial identical in the CHEmotactic (sugar-)response in *E. coli* are-in principle-well known in their mode of function, yet, important details are still missing for a detailed description of their variable effectivity.

The chosen TCS systems from bacterial phytochromes senses light and thereby provides advantages in the experimental work such as fully synchronized on- or off control as also precise locationally determined activity switch (by precisely focused light beam application). The authors follow a formerly laid path by combining sensing and signaling components from different organisms, and by variation of the individual parts a gradation in efficiency is identified and, at least, partially explained by follow-up experiments. Gene expression control by the activated RR is followed by biosynthesis of a fluorescent protein.

Remarkable differences in functionality in the cross-over combined recombinant proteins is qualitatively explained, yet, the expression system (control of promoter activity) including also

repressor components is fairly complex and does not fully allow quantification. The authors add also the differently strong phosphatase activity to their explanation.

Linker length studies follow formerly presented (heptade) work by one of the senior authors (AM) on blue light-regulated HKs. This former result is now identified also for this class of bacterio-phytochromes, in addition also supported by a sequence alignment survey in databases.

In total, the study identifies strong factors that regulate the activity of recombinant phytochrome-driven HKs resp. phosphatases (photochemical/thermal parameters), further, a structural conformational motif identified in HKs responding to other stimuli is present also in this class and a sequence alignment underpinnes this building principle.

This study is quite focused, and it is not given that the findings here-though precise up to the ultimate detail-will capture the attention of the broader interests of the community beyond the phytochrome-light-driven HKs.

Comments to individual sections:

Abstract

The introduction holds 15 lines, out of which six read as a section from a textbook. In addition, some space is given to ‘... the derivative TCSs support novel applications in synthetic biology and optogenetics’ – an aspect (application) that is not dealt with in the manuscript. To the impression of the reviewer, the authors waste precious space here.

Introduction

Following a generic description of TCSs, the bacteriophytochrome from *D. radiodurans* is introduced here quite abruptly. It might be better to have the generic part and the phy one better separated (new paragraph?)

l. 65 It is unconventional to start figures with a supplementary one (that is presewnted only in the end making the inspection quite cumbersome.

l. 84, l. 302 The arrow symbol should be change into two oppositely directed half arrows. The symbol used here is better chosen for, e.g., a rearrangement of double bonds in an aromatic ring.

l. 90 – 97 The lines present already several results in quite detail.

l. 394 The aspect of allosteric regulation is not fully convincingly presented. It includes conformational motion that on the basis of the data presented here is not conclusively shown.

Results section

The experiments are performed with great expertise and precision, as is the interpretation of the data.

Response to Reviewers

We thank the editor and the reviewers for the equally swift and careful study of our manuscript, and for the incisive and positive remarks. We enclose a revised version of our manuscript which addresses these comments as detailed below (original comments in italics, responses in blue):

Reviewer #1:

In this manuscript the authors increase the sensitivity of their recent red light responsive bacterial gene expression systems pREDawn (Red OFF) and pREDDusk (Red ON) by swapping the photosensory domain of the Deinococcus radiodurans bacteriophytochrome-derived photoreceptor BphP from which these systems are built with that of Deinococcus maricopensis, which had been identified to have robust signaling properties in a previous screen. They go on to characterize the photoresponse and phosphorylation signaling in bacteria and in vitro and convincingly show that the increase in sensitivity is due to altered signal transduction between the photosensor and phosphoryl-signaling domains. In doing so, they build and characterize a number of other variants of the engineered D. maricopensis photoreceptor kinase with different signaling and thus gene regulatory properties. In particular, they construct libraries of varied linker lengths and begin to identify principles of linker periodicity and register that play a role in kinase signaling logic. I agree with the authors that the increased light sensitivity of their new variants is useful for optogenetics research, particularly when light intensity is limiting (e.g. in vivo, potentially in a bioreactor). Overall, the paper is a valuable contribution to both the optogenetics and histidine kinase signaling literature.

We thank the reviewer for the positive verdict.

I do not understand the argument that the authors make about quantum yield not explaining the increase in sensitivity of DmRedDusk versus DrRedDusk (lines 122-126). It seems they are assuming that the Dm PSM and the DrPSM both exhibit quantum yields of 0.1 – 0.2, because that is generally the quantum yield exhibited by BphP PSMs. However, it is not clear there is a priori evidence confirming that both of these PSMs have quantum yields in this range. What if DrPSM had an unusually low quantum yield for example (e.g. 0.01)? That then may explain the difference in light sensitivity. The authors should provide more clearly justify how they can exclude this, or similar, possibilities.

The reviewer raises an important point which we followed up on. To the best of our knowledge, no pertinent experimental data on the DrPSM are available in the literature, nor on the full-length receptor. However, a recent simulation study (now cited in the revised manuscript) put the quantum yield for Pr→Pfr photoconversion at 0.15, i.e., well within the range of experimental values reported for other bacteriophytochromes. To reflect the absence of relevant experimental quantum-yield data, we rephrase the corresponding sentence in the manuscript and thereby tone down our original statement.

Moreover, further down in the manuscript (see Fig. 2c), we compare the Pr→Pfr photoconversion kinetics for DmF1 vs. DrF1 for a set red-light power and find them very similar. As stated in the manuscript, this finding points to these two bacteriophytochromes having similar quantum efficiencies for photoconversion.

What is the explanation for the bimodality in the dark state for the pDawn systems? It is presumably due to the cI inverter circuit as it is absent in the pDusk systems. The higher output

gene expression mode of the dark state overlaps with the red light activated DmRedDawn system, which reduces the utility of the system for optogenetics to a degree.

We concur that the bimodality likely owes to the continuous expression (in darkness) of the strong cI repressor and add a sentence in the manuscript to this effect. We further agree that the utility of DmREDawn is likely limited and instead recommend using the newly generated DmDERusk.

On lines 300-301, I believe the authors are referring to DmDERusk and DrDERusk rather than DmREDusk and DrREDusk.

That is indeed a mistake which we correct in the revised version. We thank the reviewer for the careful reading.

The authors should explain how the PATCHY library construction method works and the composition of the libraries created by this method as it relates to SHK engineering in the main text. Currently this is not described, making it difficult to understand why the library that was generated was generated.

We include a schematic detailing the PATCHY method as a new panel in Suppl. Fig. 7. Moreover, we expand the main text to provide salient details on the method.

The SHK nomenclature in Figure 3A is not defined, making the figure difficult to interpret. The authors should define the nomenclature in the main text and/or figure legend. What is the difference between a linker variant named by a number (e.g. 8) versus a number and the letter "a" (e.g. -18a) versus a number and the letter "b" (e.g. -18b)?

True. We apologize for this omission and include descriptions of the SHK nomenclature in the main text and in the legends to Fig. 3 and Suppl. Fig. 7.

I don't think that Figure 5 a – b adds much to the impact of the of the paper. In particular, the authors do not achieve independent control of the expression of two genes with blue and red light using these constructs, due to the fact that they cross-talk through cI repressor. I suppose that one could use this system as a blue OR red light gate, but the authors do not comment on that. I recommend they be moved to the supplementary information and replaced with schematics of the engineered genetic devices used in panels c and d..

We agree with this assessment and thus relegate panels a and b of Fig. 5 to the new Suppl. Fig. 10. In addition, we include schematics of the pAurora and DmDERusk circuits in Fig. 5, as well as those of pAurora and DrREDawn in Suppl. Fig. 10.

Reviewer #2:

The authors target one of the cornerstones in the regulation of microbial lifestyle. Histidine kinases (HKs)-response regulators (RRs), or, TCS (two component systems) are ubiquitous and (i) sense a multitude of external stimuli from the environment, and (ii) give a wide variation of intracellular signals allowing for a physiological response of the living organism. The components of these systems-often still coined as Che- (CheY, -W, -A etc.) from their initial identical in the CHEmotactic (sugar-)response in E. coli are-in principle-well known in their mode of function, yet, important details are still missing for a detailed description of their variable effectivity.

The chosen TCS systems from bacterial phytochromes senses light and thereby provides advantages in the experimental work such as fully synchronized on- or off control as also precise

locationally determined activity switch (by precisely focused light beam application). The authors follow a formerly laid path by combining sensing and signaling components from different organisms, and by variation of the individual parts a gradation in efficiency is identified and, at least, partially explained by follow-up experiments. Gene expression control by the activated RRs is followed by biosynthesis of a fluorescent protein.

Remarkable differences in functionality in the cross-over combined recombinant proteins is qualitatively explained, yet, the expression system (control of promoter activity) including also repressor components is fairly complex and does not fully allow quantification. The authors add also the differently strong phosphatase activity to their explanation.

Linker length studies follow formerly presented (heptade) work by one of the senior authors (AM) on blue light-regulated HKs. This former result is now identified also for this class of bacteriophytochromes, in addition also supported by a sequence alignment survey in databases.

In total, the study identifies strong factors that regulate the activity of recombinant phytochrome-driven HKs resp. phosphatases (photochemical/thermal parameters), further, a structural conformational motif identified in HKs responding to other stimuli is present also in this class and a sequence alignment underpins this building principle.

This study is quite focused, and it is not given that the findings here-though precise up to the ultimate detail-will capture the attention of the broader interests of the community beyond the phytochrome-light-driven HKs.

We thank the reviewer for putting our contribution into the context of two-component system (TCS) research and for the constructive comments.

As this reviewer notes, studies on TCSs are frequently complicated by several aspects, chiefly the transmembrane nature of most sensor histidine kinases (SHK) and challenges associated with precisely dosing the cognate signal that the TCS responds to. Consequently, TCSs are often mainly assessed at the level of the cellular gene-expression output, whereas quantitative biochemical characterization *in vitro* is conducted more rarely.

As noted by the reviewer, photoreceptor histidine kinases can overcome these hurdles in that they are often soluble proteins and in that light as the signal can be introduced, precisely dosed, and withdrawn with comparative ease. In our contribution, we duly exploit these advantages afforded by light-responsive TCSs. By using a bacteriophytochrome-controlled TCS as a paradigm, we analyzed the response to light signals not only at the cellular level but also in biochemical experiments. Doing so pinpointed the overarching importance of the dual kinase and phosphatase activities of SHKs and their dependence on linker properties. We thus identify a means for tuning the signal response in TCSs. Taken together, we present a comparatively extensive characterization of the paradigm TCSs and the underlying SHKs. Beyond their evident utility for biotechnological applications, our results also provide lessons for TCS signaling and mechanism in general, a sentiment shared by reviewer #1.

Comments to individual sections:

Abstract

The introduction holds 15 lines, out of which six read as a section from a textbook. In addition, some space is given to ‘... the derivative TCSs support novel applications in synthetic biology and optogenetics’ – an aspect (application) that is not dealt with in the manuscript. To the impression of the reviewer, the authors waste precious space here.

We shorten the beginning of the abstract. At the same time, we strive to properly introduce the salient aspects, i.e., the roles of the sensor histidine kinase (SHK) and the response regulator, and the kinase-phosphatase duality of SHKs.

We concur that, strictly speaking, our manuscript does not demonstrate “novel applications in synthetic biology and optogenetics”. This sentence was rather meant to indicate future prospects afforded by our work. To better reflect this aspect, we rephrase the sentence in question accordingly.

Introduction

Following a generic description of TCSs, the bacteriophytochrome from D. radiodurans is introduced here quite abruptly. It might be better to have the generic part and the phy one better separated (new paragraph?)

We agree and introduce a new paragraph when first discussing the *D. radiodurans* bacteriophytochrome.

l. 65 It is unconventional to start figures with a supplementary one (that is presented only in the end making the inspection quite cumbersome).

We concur and expand Fig. 1 in the main manuscript by a new panel **a** showing the domain architecture of bacteriophytochromes. In the revised manuscript, we refer to this new figure panel first before mentioning any supplementary figures.

l. 84, l. 302 The arrow symbol should be change into two oppositely directed half arrows. The symbol used here is better chosen for, e.g., a rearrangement of double bonds in an aromatic ring.

Agreed and done.

l. 90 – 97 The lines present already several results in quite detail.

On balance, we prefer to retain this preview of our work at the end of the Introduction section because it conveys to the reader the general scope of the study. That said, we do remove one detailed reference to a specific result which was indeed unnecessarily concrete.

l. 394 The aspect of allosteric regulation is not fully convincingly presented. It includes conformational motion that on the basis of the data presented here is not conclusively shown.

As aptly noted by the reviewer further up, “important details are still missing for a detailed description” in the characterization of two-component signaling. We wholeheartedly agree and have hence striven to critically evaluate our present data and propose a candidate mechanistic model that accounts for them. As we state in the text, this model is largely based on the linker variations observed at present, recent cryo-EM structures of the *Deinococcus radiodurans* bacteriophytochrome, and structural data on the thermosensing histidine kinase DesK from *Bacillus subtilis*. All that said, we concur that other models may apply as well and add a statement to this effect.

Results section

The experiments are performed with great expertise and precision, as is the interpretation of the data.

We appreciate the generous comment.

Beyond the above changes, we also noted and corrected an error in the linker-length analyses of natural sensor histidine kinases in Fig. 3 panels b and c. The distributions were displaced by one residue which we have now amended. The conclusions of the manuscript are unaffected by this inadvertent mistake. Moreover, we have modified and reformatted the manuscript according to the journal guidelines.

REVIEWERS' COMMENTS

Reviewer #1 (Remarks to the Author):

My concerns have been addressed.

Reviewer #2 (Remarks to the Author):

Report to 486044_1

Here, the authors present a revised version of their manuscript entitled ,Leveraging the Histidine Kinase-Phosphatase Duality to Sculpt Two-Component Signaling‘.

In this revision and in their rebuttal letter, the authors addressed all points, comments, and arguments raised by both reviewers.

It should be noted here that reviewer #1 argues to determine in greater detail the quantum yields of bacteriophytochromes, as low quantum yields may interfere with the results reported here. In fact, inspection of Consiglieri et al. (2019) PPS, lists quantum yields for several bacterial and fungal phytochromes with very low numbers. As there is overall little literature available to that parameter, so the authors may consider to consider a comparative measurement using a well determined phytochrome as standard.

Overall, the revision has solved most aspects given by the reviewers.

Response to Reviewers

We thank the editor and the reviewers for the positive verdict on our manuscript and the provisional acceptance. We address the final reviewer comments as follows (original comments in italics, responses in blue):

Reviewer #1:

My concerns have been addressed.

Reviewer #2:

Here, the authors present a revised version of their manuscript entitled ‘Leveraging the Histidine Kinase-Phosphatase Duality to Sculpt Two-Component Signaling’.

In this revision and in their rebuttal letter, the authors addressed all points, comments, and arguments raised by both reviewers.

It should be noted here that reviewer #1 argues to determine in greater detail the quantum yields of bacteriophytochromes, as low quantum yields may interfere with the results reported here. In fact, inspection of Consiglieri et al. (2019) PPS, lists quantum yields for several bacterial and fungal phytochromes with very low numbers. As there is overall little literature available to that parameter, so the authors may consider to consider a comparative measurement using a well determined phytochrome as standard.

We include a reference to the work by Consiglieri *et al.* in the manuscript. Further, we note that in response to reviewer #1’s earlier comment, we had already modified our text and toned down our original statement about the quantum yields.

Notably, for the present manuscript, the relative values of the Pr→Pfr photoconversion quantum yields in DrF1 (based on the PSM from *Deinococcus radiodurans*) vs. DmF1 (based on the PSM from *Deinococcus maricopensis*) matter, rather than their absolute values. As indicated by the experiments detailed in Fig. 2c, these quantum yields are closely similar in DrF1 and DmF1. We conclude that the quantum yields (whose absolute values are unknown at this point) do not “interfere with the results reported”.

On a general note, we appreciate this reviewer’s comment and plan to experimentally determine the Pr→Pfr photoconversion quantum yields in the future.

Overall, the revision has solved most aspects given by the reviewers.

Other than that, we have modified and reformatted the manuscript according to the journal guidelines provided to us.